# Convergence of Sharpness-Aware Minimization Algorithms using Increasing Batch Size and Decaying Learning Rate

## Abstract

The sharpness-aware minimization (SAM) algorithm and its variants, including gap guided SAM (GSAM), have been successful at improving the generalization capability of deep neural network models by finding flat local minima of the empirical loss in training. Meanwhile, it has been shown theoretically and practically that increasing the batch size or decaying the learning rate avoids sharp local minima of the empirical loss. In this paper, we consider the GSAM algorithm with increasing batch sizes or decaying learning rates, such as cosine annealing or linear learning rate, and theoretically show its convergence. Moreover, we numerically compare SAM (GSAM) with and without an increasing batch size and conclude that using an increasing batch size or decaying learning rate finds flatter local minima than using a constant batch size and learning rate.

## 1 Introduction

One way to train a deep neural network (DNN) is to find an optimal parameter $x^\star$ of the network in the sense of minimizing the empirical loss $f_S(x) = \frac{1}{n} \sum_{i \in [n]} f_i(x)$ given by the training set $S = (z_1, z_2, \cdots, z_n)$ and a nonconvex loss function $f(x; z_i) = f_i(x)$ corresponding to the $i$-th training data $z_i \in S$ ($i \in [n] := \{1, 2, \cdots, n\}$). Our main concern is whether a DNN trained by an algorithm for empirical risk minimization (ERM), wherein the empirical loss $f_S$ is minimized, has a strong generalization capability. The sharpness-aware minimization (SAM) problem (Foret et al., 2021) was proposed as a way to improve a DNN's generalization capability. The SAM problem is to minimize a perturbed empirical loss defined as the maximum empirical loss $f_{S,\rho}(x) := \max_{\|\epsilon\| \le \rho} f_S(x + \epsilon)$ over a certain neighborhood of a parameter $x \in \mathbb{R}^d$ of the DNN, where $\rho \ge 0$ and $\epsilon \in \mathbb{R}^d$. From the definition of the perturbed empirical loss $f_{S,\rho}$, the SAM problem is specialized to finding flat local minima of the empirical loss $f_S$, which may lead to a better generalization capability than finding sharp minima (Keskar et al., 2017; Jiang et al., 2020). Although (Andriushchenko et al., 2023b) reported that the relationship between sharpness and generalization would be weak, the SAM algorithm and its variants for solving the SAM problem have high generalization capabilities and superior performance, as shown in, e.g., (Chen et al., 2022; Du et al., 2022; Andriushchenko et al., 2023a; Wen et al., 2023; Chen et al., 2023; Möllenhoff & Khan, 2023; Wang et al., 2024; Sherborne et al., 2024; Springer et al., 2024).

Meanwhile, an algorithm using a large batch size falls into sharp local minima of the empirical loss $f_S$ and the algorithm would experience a drop in generalization performance (Hoffer et al., 2017; Goyal et al., 2018; You et al., 2020). It has been shown that increasing the batch size (Byrd et al., 2012; Balles et al., 2017; De et al., 2017; Smith et al., 2018; Goyal et al., 2018) or decaying the learning rate (Wu et al., 2014; Ioffe & Szegedy, 2015; Loshchilov & Hutter, 2017; Hundt et al., 2019) avoids sharp local minima of the empirical loss. Hence, we are interested in verifying whether the SAM algorithm with an increasing batch size or decaying learning rate performs well in training DNNs. In this paper, we focus on the SAM algorithm called gap guided SAM (GSAM) algorithm (Zhuang et al., 2022) (see Algorithm 1 for details).

**Contribution:** The main contribution of this paper is to show an $\epsilon$-approximation of the GSAM algorithm **with an increasing batch size and constant learning rate** ((7) in Table 1; Theorem 2.3) and **with a constant batch size and decaying learning rate** ((8) in Table 1; Theorem 2.4).

Table 1: Convergence of SAM and its variants to minimize $\hat{f}_{S,\rho}^{\mathrm{SAM}}(\boldsymbol{x}) = f_S(\boldsymbol{x}) + \rho\|\nabla f_S(\boldsymbol{x})\|$ over the number of steps $T$. "Noise" in the Gradient column means that algorithm uses noisy observation, i.e., $\boldsymbol{g}(\boldsymbol{x}) = \nabla f(\boldsymbol{x}) + (\text{Noise})$, of the full gradient $\nabla f(\boldsymbol{x})$, while "Mini-batch" in the Gradient column means that algorithm uses a mini-batch gradient $\nabla f_B(\boldsymbol{x}) = \frac{1}{b}\sum_{i \in [b]} \nabla f_{\xi_i}(\boldsymbol{x})$ with a batch size $b$. Here, we let $\mathbb{E}[\|\nabla \hat{f}_{S,\rho}^{\mathrm{SAM}*}\|] := \min_{t \in [T]} \mathbb{E}[\|\nabla \hat{f}_{S,\rho}^{\mathrm{SAM}}(\boldsymbol{x}_t)\|]$, where $(\boldsymbol{x}_t)_{t=0}^T$ is the sequence generated by Algorithm. Results (1)–(6) were presented in (1) (Andriushchenko & Flammarion, 2022, Theorem 2), (2) (Mi et al., 2022, Theorem 2), (3) (Zhuang et al., 2022, Theorem 5.1), (4) (Si & Yun, 2023, Theorem 4.6), (5) (Li & Giannakis, 2023, Corollary 1), and (6) (Li et al., 2024, Theorem 2).

| Algorithm | Gradient | Leaning Rate | Perturbation | Convergence Analysis |
|---|---|---|---|---|
| (1) SAM | Mini-batch $b$ | $\eta_T = \Theta(\frac{1}{T^{1/2}})$ | $\rho_T = \Theta(\frac{1}{T^{1/4}})$ | $\mathbb{E}[\|\nabla f_S^*\|] = O(\frac{1}{T^{1/4}} + \frac{1}{bT^{1/4}})$ |
| (2) SSAM | Noise | $\eta_t = \Theta(\frac{1}{t^{1/2}})$ | $\rho_t = \Theta(\frac{1}{t^{1/2}})$ | $\mathbb{E}[\|\nabla f_S^*\|] = O(\frac{\sqrt{\log T}}{T^{1/4}})$ |
| (3) GSAM | Noise | $\eta_t = \Theta(\frac{1}{t^{1/2}})$ | $\rho_t = \Theta(\frac{1}{t^{1/2}})$ | $\mathbb{E}[\|\nabla \hat{f}_{S,\rho_t}^{\mathrm{SAM}*}\|] = O\left(\frac{\sqrt{\log T}}{T^{1/4}}\right)$ |
| (4) $m$-SAM | Noise | $\eta_T = O(\frac{1}{T^{1/2}})$ | $\rho$ | $\mathbb{E}[\|\nabla f_S^*\|] = O(\sqrt{\frac{1}{T^{1/2}} + \rho^2})$ |
| (5) VaSSO | Noise | $\eta_T = \Theta(\frac{1}{T^{1/2}})$ | $\rho_T = \Theta(\frac{1}{T^{1/2}})$ | $\mathbb{E}[\|\nabla \hat{f}_{S,\rho}^{\mathrm{SAM}*}\|] = O(\frac{1}{T^{1/4}})$ |
| (6) FSAM | Noise | $\eta_T = \Theta(\frac{1}{T^{1/2}})$ | $\rho_t = \Theta(\frac{1}{t^{1/2}})$ | $\mathbb{E}[\|\nabla f_S^*\|] = O(\frac{\sqrt{\log T}}{T^{1/4}})$ |
| (7) GSAM **[Ours]** | Increasing mini-batch $b_t$ | Constant $\eta = O(n\epsilon^2)$ | $\rho = O(\frac{nb_0\epsilon^2}{\sqrt{n^2+b_0^2}})$ | $\mathbb{E}[\|\nabla \hat{f}_{S,\rho}^{\mathrm{SAM}*}\|] \le \epsilon$ |
| (8) GSAM **[Ours]** | Mini-batch $b$ | Cosine/Linear $\eta_t \to \underline{\eta}\ (\ge 0)$ | $\rho = O(\frac{nb\epsilon^2}{\sqrt{n^2+b^2}})$ | $\mathbb{E}[\|\nabla \hat{f}_{S,\rho}^{\mathrm{SAM}*}\|] \le \epsilon$ |

Our convergence analyses of GSAM are based on the search direction noise $\eta_t\boldsymbol{\omega}_t$ (defined by (9)) between GSAM and gradient descent (GD) (Theorems 2.1 and 2.2 in Section 2.3). The norm of the noise is approximately $\Theta(\frac{\eta_t}{\sqrt{b_t}})$ (see also (10)). Since this implies that GSAM using a large batch size $b$ or a small learning rate $\eta$ behaves approximately the same as GD in solving the SAM problem, GSAM eventually needs to use a large batch size or a small learning rate. Accordingly, it will be useful to use increasing batch sizes or decaying learning rates, as the previous results presented in the second paragraph of this section point out. We would also like to emphasize that our analyses allow us to use practical learning rates, such as constant, cosine-annealing, and linear learning rates, unlike the existing methods listed in Table 1. Our other contribution is to provide numerical results on training ResNets and ViT-Tiny on the CIFAR100 dataset such that using a doubly increasing batch size or a cosine-annealing learning rate finds flatter local minima than using a constant batch size and learning rate (Section 3 and Appendix C).

**Related work:** Convergence analyses of SGD (Robbins & Monro, 1951) with a fixed batch size have been presented in (Ghadimi & Lan, 2013; Ghadimi et al., 2016; Vaswani et al., 2019; Fehrman et al., 2020; Chen et al., 2020; Scaman & Malherbe, 2020; Loizou et al., 2021; Wang et al., 2021; Arjevani et al., 2023; Khaled & Richtárik, 2023). Our analyses found that SGD (an example of GSAM) using increasing batch sizes or a cosine-annealing (linear) learning rate is an $\epsilon$-approximation. The linear scaling rule (Goyal et al., 2018; Smith et al., 2018; Xie et al., 2021) based on $\frac{\eta}{b}$ coincides with our rule based on the noise norm $\eta\|\boldsymbol{\omega}_t\|^2 = \Theta(\frac{\eta_t}{b_t})$. In (Hazan et al., 2016; Sato & Iiduka, 2023), it was shown that SGD with an increasing batch size reaches the global optimum under the strong convexity assumption of the smoothed function of $f_S$. This paper shows that, with nonconvex loss functions, GSAM with an increasing batch size achieves an $\epsilon$-approximation.

**Limitations:** The limitation of this study is the limited number of models and datasets used in the experiments. Hence, we should conduct similar experiments with a larger number of models and datasets to support our theoretical results.

## 2 SAM PROBLEM AND GSAM

Let $\mathbb{N}$ be the set of natural numbers. Let $[n] := \{1, 2, \cdots, n\}$ and $[0 : n] := \{0, 1, \cdots, n\}$ for $n \in \mathbb{N}$. Let $\mathbb{R}^d$ be a $d$-dimensional Euclidean space with inner product $\langle \boldsymbol{x}, \boldsymbol{y} \rangle_2 = \boldsymbol{x}^\top \boldsymbol{y}$ $(\boldsymbol{x}, \boldsymbol{y} \in \mathbb{R}^d)$ and its induced norm $\|\boldsymbol{x}\|_2 := \sqrt{\langle \boldsymbol{x}, \boldsymbol{x} \rangle_2}$ $(\boldsymbol{x} \in \mathbb{R}^d)$. The gradient and Hessian of a twice differentiable function $f \colon \mathbb{R}^d \to \mathbb{R}$ at $\boldsymbol{x} \in \mathbb{R}^d$ are denoted by $\nabla f(\boldsymbol{x})$ and $\nabla^2 f(\boldsymbol{x})$, respectively. Let $L > 0$. A differentiable function $f \colon \mathbb{R}^d \to \mathbb{R}$ is said to be $L$–smooth if the gradient $\nabla f \colon \mathbb{R}^d \to \mathbb{R}^d$ is Lipschitz continuous; i.e., for all $\boldsymbol{x}, \boldsymbol{y} \in \mathbb{R}^d$, $\|\nabla f(\boldsymbol{x}) - \nabla f(\boldsymbol{y})\|_2 \leq L\|\boldsymbol{x} - \boldsymbol{y}\|_2$. Let $O$ and $\Theta$ be Landau's symbols, i.e., $y_t = O(x_t)$ (resp. $y_t = \Theta(x_t)$) if there exist $c > 0$ (resp. $c_1, c_2 > 0$) and $t_0 \in \mathbb{N}$ such that, for all $t \geq t_0$, $y_t \leq cx_t$ (resp. $c_1 x_t \leq y_t \leq c_2 x_t$).

### 2.1 SAM PROBLEM AND ITS APPROXIMATION PROBLEM

Given a parameter $\boldsymbol{x} \in \mathbb{R}^d$ and a data point $z$, a machine-learning model provides a prediction whose quality can be measured by a differentiable nonconvex loss function $f(\boldsymbol{x}; z)$. For a training set $S = (z_1, z_2, \ldots, z_n)$, $f_i(\cdot) := f(\cdot; z_i)$ is the loss function corresponding to the $i$-th training data $z_i$. The empirical risk minimization (ERM) is to minimize the empirical loss defined for all $\boldsymbol{x} \in \mathbb{R}^d$ by

$$f_S(\boldsymbol{x}) = \frac{1}{n} \sum_{i \in [n]} f(\boldsymbol{x}; z_i) = \frac{1}{n} \sum_{i \in [n]} f_i(\boldsymbol{x}). \tag{1}$$

Given $\rho \geq 0$ and a training set $S$, the SAM problem (Foret et al., 2021, (1)) is to minimize

$$f_{S,\rho}^{\mathrm{SAM}}(\boldsymbol{x}) := \max_{\|\boldsymbol{\epsilon}\|_2 \leq \rho} f_S(\boldsymbol{x} + \boldsymbol{\epsilon}). \tag{2}$$

Let $\boldsymbol{x} \in \mathbb{R}^d$ and $\rho \geq 0$. Taylor's theorem thus implies that there exists $\tau = \tau(\boldsymbol{x}, \rho) \in (0, 1)$ such that the maximizer $\boldsymbol{\epsilon}_{S,\rho}^\star(\boldsymbol{x})$ of $f_S(\boldsymbol{x} + \boldsymbol{\epsilon})$ over $B_2(\boldsymbol{0}; \rho) := \{\boldsymbol{\epsilon} \in \mathbb{R}^d \colon \|\boldsymbol{\epsilon}\|_2 \leq \rho\}$ is as follows:

$$\boldsymbol{\epsilon}_{S,\rho}^\star(\boldsymbol{x}) := \underset{\|\boldsymbol{\epsilon}\|_2 \leq \rho}{\arg\max}\, f_S(\boldsymbol{x} + \boldsymbol{\epsilon}) = \underset{\|\boldsymbol{\epsilon}\|_2 \leq \rho}{\arg\max} \left\{ f_S(\boldsymbol{x}) + \langle \nabla f_S(\boldsymbol{x}), \boldsymbol{\epsilon} \rangle_2 + \frac{1}{2} \langle \boldsymbol{\epsilon}, \nabla^2 f_S(\boldsymbol{x} + \tau\boldsymbol{\epsilon})\boldsymbol{\epsilon} \rangle_2 \right\},$$

where we suppose that $f_S$ is twice differentiable on $\mathbb{R}^d$. Then, assuming $\|\boldsymbol{\epsilon}\|_2^2 \approx 0$ (i.e., a small enough $\rho^2$), $\boldsymbol{\epsilon}_{S,\rho}^\star(\boldsymbol{x})$ can be approximated as follows $\hat{\boldsymbol{\epsilon}}_{S,\rho}(\boldsymbol{x})$ (Foret et al., 2021, (2)):

$$\boldsymbol{\epsilon}_{S,\rho}^\star(\boldsymbol{x}) \approx \hat{\boldsymbol{\epsilon}}_{S,\rho}(\boldsymbol{x}) := \underset{\|\boldsymbol{\epsilon}\|_2 \leq \rho}{\arg\max} \langle \nabla f_S(\boldsymbol{x}), \boldsymbol{\epsilon} \rangle_2 = \begin{cases} \left\{ \rho \frac{\nabla f_S(\boldsymbol{x})}{\|\nabla f_S(\boldsymbol{x})\|_2} \right\} & (\nabla f_S(\boldsymbol{x}) \neq \boldsymbol{0}) \\ B_2(\boldsymbol{0}; \rho) & (\nabla f_S(\boldsymbol{x}) = \boldsymbol{0}). \end{cases} \tag{3}$$

Here, our goal is to solve the following problem that is an approximation of the SAM problem of minimizing $f_{S,\rho}^{\mathrm{SAM}}(\boldsymbol{x}) = \max_{\|\boldsymbol{\epsilon}\|_2 \leq \rho} f_S(\boldsymbol{x} + \boldsymbol{\epsilon})$ (see (2) and (3)).

**Problem 2.1 (Approximated SAM problem (Foret et al., 2021))** *Let $f_S$ be the empirical loss defined by (1) with the training set $S = (z_1, z_2, \cdots, z_n)$. Given $\rho \geq 0$,*

*minimize $\hat{f}_{S,\rho}^{\mathrm{SAM}}(\boldsymbol{x}) := \max_{\|\boldsymbol{\epsilon}\|_2 \leq \rho} \{f_S(\boldsymbol{x}) + \langle \nabla f_S(\boldsymbol{x}), \boldsymbol{\epsilon} \rangle_2\} = f_S(\boldsymbol{x}) + \rho\|\nabla f_S(\boldsymbol{x})\|_2$ subject to $\boldsymbol{x} \in \mathbb{R}^d$.*

We use the following approximation (Foret et al., 2021, (3)) of the gradient of $\hat{f}_{S,\rho}^{\mathrm{SAM}}$ at $\boldsymbol{x} \in \mathbb{R}^d$:

$$\nabla \hat{f}_{S,\rho}^{\mathrm{SAM}}(\boldsymbol{x}) := \nabla f_S(\boldsymbol{x})|_{\boldsymbol{x} + \hat{\boldsymbol{\epsilon}}_{S,\rho}(\boldsymbol{x})} = \begin{cases} \nabla f_S \left( \boldsymbol{x} + \rho \frac{\nabla f_S(\boldsymbol{x})}{\|\nabla f_S(\boldsymbol{x})\|_2} \right) & (\nabla f_S(\boldsymbol{x}) \neq \boldsymbol{0}) \\ \nabla f_S(\boldsymbol{x} + \boldsymbol{u}) & (\nabla f_S(\boldsymbol{x}) = \boldsymbol{0}), \end{cases} \tag{4}$$

where $\hat{\boldsymbol{\epsilon}}_{S,\rho}(\boldsymbol{x})$ is denoted by (3) and $\boldsymbol{u}$ is an arbitrary point in $B_2(\boldsymbol{0}; \rho)$ (e.g., we may set $\boldsymbol{u} = \boldsymbol{0}$ before implementing algorithms).

### 2.2 MINI-BATCH GSAM ALGORITHM

As a way of solving Problem 2.1, we will study the GSAM algorithm (Zhuang et al., 2022, Algorithm 1) using $b$ loss functions $f_{\xi_{t,1}}, f_{\xi_{t,2}}, \cdots, f_{\xi_{t,b}} \in \{f_1, f_2, \cdots, f_n\}$ which are randomly chosen at each time $t$, where $b$ is a batch size satisfying $b \leq n$. We suppose that loss functions satisfy the following conditions.

**Assumption 2.1** (A1) $f_i \colon \mathbb{R}^d \to \mathbb{R}$ ($i \in [n]$) *is twice differentiable and $L_i$-smooth.*

(A2) $\nabla f_\xi \colon \mathbb{R}^d \to \mathbb{R}^d$ *is the stochastic gradient of $\nabla f_S$; i.e., (i) for all $\boldsymbol{x} \in \mathbb{R}^d$, $\mathbb{E}_\xi[\nabla f_\xi(\boldsymbol{x})] = \nabla f_S(\boldsymbol{x})$, (ii) there exists $\sigma \geq 0$ such that, for all $\boldsymbol{x} \in \mathbb{R}^d$, $\mathbb{V}_\xi[\nabla f_\xi(\boldsymbol{x})] = \mathbb{E}_\xi[\|\nabla f_\xi(\boldsymbol{x}) - \nabla f_S(\boldsymbol{x})\|_2^2] \leq \sigma^2$, where $\xi$ is a random variable which is independent of $\boldsymbol{x}$ and $\mathbb{E}_\xi[\cdot]$ stands for the expectation with respect to $\xi$.*

(A3) *Let $t \in \mathbb{N}$ and suppose that $b_t \in \mathbb{N}$ and $b_t \leq n$. Let $\boldsymbol{\xi}_t = (\xi_{t,1}, \xi_{t,2}, \cdots, \xi_{t,b_t})^\top$ be a random variable that consists of $b_t$ independent and identically distributed variables. The full gradient $\nabla f_S(\boldsymbol{x})$ is estimated as the following mini-batch gradient at $\boldsymbol{x} \in \mathbb{R}^d$:*

$$\nabla f_{S_t}(\boldsymbol{x}) := \frac{1}{b_t} \sum_{i \in [b_t]} \nabla f_{\xi_{t,i}}(\boldsymbol{x}), \tag{5}$$

*where $\boldsymbol{\xi}_t$ is independent of $\boldsymbol{x}$, $b_t$, and $\boldsymbol{\xi}_{t'}$ ($t \neq t'$).*

We define $\hat{\epsilon}_{S_t,\rho}$ by replacing $S$ in (3) with $S_t$ in (A3), i.e.,

$$\hat{\boldsymbol{\epsilon}}_{S_t,\rho}(\boldsymbol{x}) := \arg\max_{\|\boldsymbol{\epsilon}\|_2 \leq \rho} \langle \nabla f_{S_t}(\boldsymbol{x}), \boldsymbol{\epsilon} \rangle_2 = \begin{cases} \left\{ \rho \frac{\nabla f_{S_t}(\boldsymbol{x})}{\|\nabla f_{S_t}(\boldsymbol{x})\|_2} \right\} & (\nabla f_{S_t}(\boldsymbol{x}) \neq \boldsymbol{0}) \\ B_2(\boldsymbol{0}; \rho) & (\nabla f_{S_t}(\boldsymbol{x}) = \boldsymbol{0}), \end{cases} \tag{6}$$

where $\nabla f_{S_t}$ is defined as in (5). Accordingly, an approximation of a mini-batch gradient of $\hat{f}_{S,\rho}^{\mathrm{SAM}}$ (see Problem 2.1 and (4)) at $\boldsymbol{x} \in \mathbb{R}^d$ can be defined as

$$\nabla \hat{f}_{S_t,\rho}^{\mathrm{SAM}}(\boldsymbol{x}) := \nabla f_{S_t}(\boldsymbol{x})|_{\boldsymbol{x}+\hat{\epsilon}_{S_t,\rho}(\boldsymbol{x})} = \begin{cases} \nabla f_{S_t}\left(\boldsymbol{x} + \rho \frac{\nabla f_{S_t}(\boldsymbol{x})}{\|\nabla f_{S_t}(\boldsymbol{x})\|_2}\right) & (\nabla f_{S_t}(\boldsymbol{x}) \neq \boldsymbol{0}) \\ \nabla f_{S_t}(\boldsymbol{x} + \boldsymbol{u}) & (\nabla f_{S_t}(\boldsymbol{x}) = \boldsymbol{0}), \end{cases} \tag{7}$$

where $\hat{\epsilon}_{S_t,\rho}(\boldsymbol{x})$ is denoted by (6) and $\boldsymbol{u}$ is an arbitrary point in $B_2(\boldsymbol{0}; \rho)$. Accordingly, the SAM algorithm (Foret et al., 2021, Algorithm 1) can be obtained by applying SGD to the objective function $\hat{f}_{S,\rho}^{\mathrm{SAM}}$ in Problem 2.1, as described in Algorithm 1. GD for Problem 2.1 coincides with Algorithm 1 with $S_t = S$ (i.e., $b_t = n$), as follows:

$$\boldsymbol{x}_{t+1} := \boldsymbol{x}_t - \eta_t \nabla \hat{f}_{S,\rho}^{\mathrm{SAM}}(\boldsymbol{x}_t), \tag{8}$$

where $\nabla \hat{f}_{S,\rho}^{\mathrm{SAM}}$ is defined as in (4). The GSAM algorithm uses an ascent step in the orthogonal direction that is obtained by using stochastic gradient decomposition $\nabla f_{S_t}(\boldsymbol{x}) = \nabla f_{S_t\|}(\boldsymbol{x}) + \nabla f_{S_t\perp}(\boldsymbol{x})$ to minimize a surrogate gap $h_t(\boldsymbol{x}) := \hat{f}_{S_t,\rho}^{\mathrm{SAM}}(\boldsymbol{x}) - f_{S_t}(\boldsymbol{x})$ (see (Zhuang et al., 2022, Section 4)).

---

**Algorithm 1** Mini-batch GSAM algorithm

---

**Require:** $\rho \geq 0$ (hyperparameter), $\boldsymbol{u} \in B_2(\boldsymbol{0}; \rho)$, $\boldsymbol{x}_0 \in \mathbb{R}^d$ (initial point), $b_t > 0$ (batch size), $\eta_t > 0$ (learning rate), $\alpha \in \mathbb{R}$ (control parameter of ascent step), $T \geq 1$ (steps)
**Ensure:** $(\boldsymbol{x}_t)_{t=0}^T \subset \mathbb{R}^d$
  **for** $t = 0, 1, \ldots, T - 1$ **do**

$$\nabla \hat{f}_{S_t,\rho}^{\mathrm{SAM}}(\boldsymbol{x}_t) := \begin{cases} \nabla f_{S_t}\left(\boldsymbol{x}_t + \rho \frac{\nabla f_{S_t}(\boldsymbol{x}_t)}{\|\nabla f_{S_t}(\boldsymbol{x}_t)\|_2}\right) & (\nabla f_{S_t}(\boldsymbol{x}_t) \neq \boldsymbol{0}) \\ \nabla f_{S_t}(\boldsymbol{x}_t + \boldsymbol{u}) & (\nabla f_{S_t}(\boldsymbol{x}_t) = \boldsymbol{0}) \end{cases} \triangleleft \text{See (5) for } \nabla f_{S_t}$$

$$\boldsymbol{d}_t := \begin{cases} -(\nabla \hat{f}_{S_t,\rho}^{\mathrm{SAM}}(\boldsymbol{x}_t) - \alpha \nabla f_{S_t\perp}(\boldsymbol{x}_t)) & (\text{GSAM}) \\ -\nabla \hat{f}_{S_t,\rho}^{\mathrm{SAM}}(\boldsymbol{x}_t) & (\text{SAM}; \alpha = 0) \\ -\nabla \hat{f}_{S_t,0}^{\mathrm{SAM}}(\boldsymbol{x}_t) = -\nabla f_{S_t}(\boldsymbol{x}_t) & (\text{SGD}; \alpha = \rho = 0) \end{cases}$$

  $\boldsymbol{x}_{t+1} := \boldsymbol{x}_t + \eta_t \boldsymbol{d}_t$
  **end for**

---

### 2.3 SEARCH DIRECTION NOISE BETWEEN GSAM AND GD

GSAM can find local minima of Problem 2.1 (by using $-\nabla \hat{f}_{S_t,\rho}^{\mathrm{SAM}}(\boldsymbol{x}_t)$) that are flatter local minima of $f_S$ (by using $\alpha \nabla f_{S_t\perp}(\boldsymbol{x}_t)$) (see (Zhuang et al., 2022, Section 4) for details). Meanwhile, GD defined as (8) (i.e., GSAM with $b_t = n$ and $\alpha = 0$) is the simplest algorithm for solving Problem 2.1.

Although this GD can minimize $\hat{f}_{S,\rho}^{\mathrm{SAM}}$ by using the full gradient $\nabla \hat{f}_{S,\rho}^{\mathrm{SAM}}(\boldsymbol{x}_t)$, it is not guaranteed that it converges to a flatter minimum of $f_S$ compared with the one of GSAM. Here, let us compare GSAM with GD. Let $\boldsymbol{x}_t \in \mathbb{R}^d$ be the $t$-th approximation of Problem 2.1 and $\eta_t > 0$. The $\boldsymbol{x}_{t+1}$ generated by GSAM is as follows:

$$
\begin{aligned}
\boldsymbol{x}_{t+1} &= \boldsymbol{x}_t + \eta_t\{-(\nabla \hat{f}_{S_t,\rho}^{\mathrm{SAM}}(\boldsymbol{x}_t) - \alpha \nabla f_{S_t\perp}(\boldsymbol{x}_t))\} \\
&= \underbrace{\boldsymbol{x}_t - \eta_t \nabla \hat{f}_{S,\rho}^{\mathrm{SAM}}(\boldsymbol{x}_t)}_{\text{GD}} + \underbrace{\eta_t(\overbrace{\nabla \hat{f}_{S,\rho}^{\mathrm{SAM}}(\boldsymbol{x}_t) - \nabla \hat{f}_{S_t,\rho}^{\mathrm{SAM}}(\boldsymbol{x}_t)}^{\hat{\boldsymbol{\omega}}_t} + \alpha \nabla f_{S_t\perp}(\boldsymbol{x}_t))}_{\text{Search Direction Noise } \eta_t \boldsymbol{\omega}_t}
\end{aligned}
\tag{9}
$$

This implies that, if $\eta_t \boldsymbol{\omega}_t := \eta_t(\nabla \hat{f}_{S,\rho}^{\mathrm{SAM}}(\boldsymbol{x}_t) - \nabla \hat{f}_{S_t,\rho}^{\mathrm{SAM}}(\boldsymbol{x}_t) + \alpha \nabla f_{S_t\perp}(\boldsymbol{x}_t))$ is approximately zero, i.e., $b_t \approx n$ and $\alpha \approx 0$, then GSAM is approximately GD in the sense of the norm of $\mathbb{R}^d$, and if $\eta_t \boldsymbol{\omega}_t$ is not zero under $\alpha \neq 0$, i.e., $b_t < n$, then the behavior of GSAM with $b_t < n$ differs from the one of GD. We call $\eta_t \boldsymbol{\omega}_t$ the *search direction noise* of GSAM, since $\eta_t \boldsymbol{\omega}_t$ is noise from the viewpoint of the search direction of GD. We provide an upper bound of the norm of the search direction noise of GSAM. Theorem 2.1 is proved in Appendix A.

**Theorem 2.1 (Upper bound of $\mathbb{E}\eta_t \|\boldsymbol{\omega}_t\|_2$)** *Suppose that Assumption 2.1 holds and define $\boldsymbol{\omega}_t \in \mathbb{R}^d$ for all $t \in \mathbb{N} \cup \{0\}$ by $\boldsymbol{\omega}_t := \hat{\boldsymbol{\omega}}_t + \alpha \nabla f_{S_t\perp}(\boldsymbol{x}_t)$, where $\boldsymbol{x}_t$ is generated by Algorithm 1 and we assume that $G_\perp := \sup_{t \in \mathbb{N} \cup \{0\}} \|\nabla f_{S_t\perp}(\boldsymbol{x}_t)\|_2 < +\infty$. Then, for all $t \in \mathbb{N} \cup \{0\}$,*

$$
\mathbb{E}[\eta_t \|\boldsymbol{\omega}_t\|_2] \leq \begin{cases} \eta_t |\alpha| G_\perp & (b_t = n) \\ \eta_t \left\{ \sqrt{4\rho^2 \left(\frac{1}{b_t^2} + \frac{1}{n^2}\right) \left(\sum_{i \in [n]} L_i\right)^2 + \frac{2\sigma^2}{b_t}} + |\alpha| G_\perp \right\} & (b_t < n), \end{cases}
$$

*where $\mathbb{E}[\cdot]$ stands for the total expectation defined by $\mathbb{E} = \mathbb{E}_{\boldsymbol{\xi}_0} \mathbb{E}_{\boldsymbol{\xi}_1} \cdots \mathbb{E}_{\boldsymbol{\xi}_t}$.*

In the case of GSAM with $b_t = n$ and $\alpha \neq 0$, we have that $\eta_t \boldsymbol{\omega}_t = \eta_t(\nabla \hat{f}_{S,\rho}^{\mathrm{SAM}}(\boldsymbol{x}_t) - \nabla \hat{f}_{S,\rho}^{\mathrm{SAM}}(\boldsymbol{x}_t) + \alpha \nabla f_{S\perp}(\boldsymbol{x}_t)) = \eta_t \alpha \nabla f_{S\perp}(\boldsymbol{x}_t)$. Hence, an upper bound of $\mathbb{E}[\eta_t \|\boldsymbol{\omega}_t\|_2]$ is $\eta_t |\alpha| G_\perp$ (Theorem 2.1 $(b_t = n)$). For simplicity, let us consider the case of $\alpha = 0$. The search direction noise $\eta_t \boldsymbol{\omega}_t$ of GSAM with $b_t < n$ is not zero, from $\nabla \hat{f}_{S,\rho}^{\mathrm{SAM}}(\boldsymbol{x}_t) \neq \nabla \hat{f}_{S_t,\rho}^{\mathrm{SAM}}(\boldsymbol{x}_t)$ (see (9)). Meanwhile, the search direction noise $\eta_t \boldsymbol{\omega}_t$ of GD (GSAM with $b_t = n$ and $\alpha = 0$) is $\eta_t \boldsymbol{\omega}_t = \eta_t(\nabla \hat{f}_{S,\rho}^{\mathrm{SAM}}(\boldsymbol{x}_t) - \nabla \hat{f}_{S,\rho}^{\mathrm{SAM}}(\boldsymbol{x}_t)) = \boldsymbol{0}$, which implies that $\mathbb{E}[\eta_t \|\boldsymbol{\omega}_t\|_2] = 0$ (This result coincides with Theorem 2.1 $(b_t = n$ and $\alpha = 0)$). Accordingly, the noise norm $\mathbb{E}[\eta_t \|\boldsymbol{\omega}_t\|_2]$ of GSAM will decrease as the batch size $b_t$ increases. In fact, from Theorem 2.1 $(b_t < n)$, the upper bound $U(\eta_t, b_t)$ of $\mathbb{E}[\eta_t \|\boldsymbol{\omega}_t\|_2]$

$$
\mathbb{E}[\eta_t \|\boldsymbol{\omega}_t\|_2] \leq \eta_t \sqrt{4\rho^2 \left(\frac{1}{b_t^2} + \frac{1}{n^2}\right) \left(\sum_{i \in [n]} L_i\right)^2 + \frac{2\sigma^2}{b_t}} \leq \eta_t \frac{\sqrt{8\rho^2(\sum_{i \in [n]} L_i)^2 + 2\sigma^2}}{\sqrt{b_t}} =: U(\eta_t, b_t)
$$

is a monotone decreasing function of $b_t$. As a result, $\mathbb{E}[\eta_t \|\boldsymbol{\omega}_t\|_2]$ decreases as $b_t$ increases. Theorem 2.1 also indicates that the smaller $\eta_t$ is, the smaller $\mathbb{E}[\eta_t \|\boldsymbol{\omega}_t\|_2]$ becomes.

Next, we provide a lower bound of the norm of the search direction noise of GSAM. Theorem 2.2 is proven in Appendix A.

**Theorem 2.2 (Lower bound of $\mathbb{E}\eta_t \|\boldsymbol{\omega}_t\|_2$)** *Under the assumptions in Theorem 2.1, for all $t \in \mathbb{N} \cup \{0\}$,*

$$
\mathbb{E}[\eta_t \|\boldsymbol{\omega}_t\|_2] \geq \begin{cases} \eta_t |\alpha| \mathbb{E}[\|\nabla f_{S\perp}(\boldsymbol{x}_t)\|_2] & (b_t = n) \\ \eta_t \left\{ \frac{c_t \sigma}{\sqrt{b_t}} - \rho\left(\frac{1}{b_t} + \frac{1}{n}\right) \sum_{i \in [n]} L_i - |\alpha| G_\perp \right\} & (b_t < n \wedge A_t \geq 0) \\ \eta_t \left\{ \rho\left(\frac{d_t}{b_t} - \frac{1}{n}\right) \sum_{i \in [n]} L_i - \frac{\sigma}{\sqrt{b_t}} - |\alpha| G_\perp \right\} & (b_t < n \wedge A_t < 0) \end{cases}
$$

*where $A_t$ is defined by (25), $c_t, d_t \in (0, 1]$, and $|\alpha|$ is small such that, for $b_t < n$, $|\alpha| \|\nabla f_{S_t\perp}(\boldsymbol{x}_t)\|_2 \leq \|\hat{\boldsymbol{\omega}}_t\|_2$.*

From the definition (9) of the search direction noise, the noise norm $\mathbb{E}[\eta_t \|\boldsymbol{\omega}_t\|_2]$ of GSAM will increase as the batch size $b_t$ decreases. We can verify this fact from Theorem 2.2 $(b_t < n \wedge A_t \geq 0)$.

For simplicity, let us consider the case where $\alpha = 0$. We set $T \geq 1$, $c := \min_{t \in [0:T]} c_t$, and $\rho \leq \frac{c\sigma}{2\sum_{i \in [n]} L_i}$ (this setting implies that $\rho$, which is used in the definition of Problem 2.1, will be a small parameter (see also (3))). Then, the lower bound $L(\eta_t, b_t)$ of $\mathbb{E}[\eta_t \|\boldsymbol{\omega}_t\|_2]$ satisfies

$$\mathbb{E}[\eta_t \|\boldsymbol{\omega}_t\|_2] \geq \eta_t \left\{ \frac{c_t \sigma}{\sqrt{b_t}} - \rho \left( \frac{1}{b_t} + \frac{1}{n} \right) \sum_{i \in [n]} L_i \right\} \geq \eta_t \frac{c_t \sigma - 2\rho \sum_{i \in [n]} L_i}{\sqrt{b_t}} =: L(\eta_t, b_t) \ (\geq 0),$$

which implies that the smaller $b_t$ is, the larger the lower bound $L(\eta_t, b_t)$ of $\mathbb{E}[\eta_t \|\boldsymbol{\omega}_t\|_2]$ becomes (We can verify this result from Theorem 2.2 ($b_t < n \wedge A_t < 0$)). Therefore, $\mathbb{E}[\eta_t \|\boldsymbol{\omega}_t\|_2]$ increases as $b_t$ decreases.

To solve Problem 2.1, we consider a mini-batch scheduler and a learning rate scheduler based on Theorems 2.1 and 2.2. To apply not only GSAM but also SAM ($\alpha = 0$) to Problem 2.1, we will assume that $|\alpha|$ is approximately zero. Theorems 2.1 and 2.2 (see also the definitions of $U(\eta_t, b_t)$ and $L(\eta_t, b_t)$) indicate that, for a given small $\rho$ and for all $t \in \mathbb{N} \cup \{0\}$,

$$\mathbb{E}[\eta_t \|\boldsymbol{\omega}_t\|_2] \approx \mathbb{E}\left[ \eta_t \left\| \nabla \hat{f}_{S,\rho}^{\mathrm{SAM}}(\boldsymbol{x}_t) - \nabla \hat{f}_{S_t,\rho}^{\mathrm{SAM}}(\boldsymbol{x}_t) \right\|_2 \right] \approx \begin{cases} \Theta \left( \frac{\eta_t}{\sqrt{b_t}} \right) & (b_t < n) \\ 0 & (b_t = n). \end{cases} \quad (10)$$

Equation (10) indicates that the full gradient $\nabla \hat{f}_{S,\rho}^{\mathrm{SAM}}(\boldsymbol{x}_0)$ substantially differs from $\nabla \hat{f}_{S_0,\rho}^{\mathrm{SAM}}(\boldsymbol{x}_0)$ with a small batch size $b_0$ or a large learning rate $\eta_0$. Meanwhile, GSAM eventually needs to use a large batch size $b$ or a small learning rate $\eta$, since the behavior of GSAM using a large $b$ or small $\eta$ is approximately like that of GD in minimizing $\hat{f}_{S,\rho}^{\mathrm{SAM}}$. Accordingly, in the process of training DNN, it would be useful to use increasing batch sizes or decaying learning rates.

## 2.4 CONVERGENCE ANALYSIS OF GSAM

### 2.4.1 INCREASING BATCH SIZE AND CONSTANT LEARNING RATE

Motivated by (Smith et al., 2018), we focus on using a constant learning rate defined for all $t \in \mathbb{N} \cup \{0\}$ by $\eta_t = \eta \in (0, +\infty)$ and a mini-batch scheduler that gradually increases the batch size:

$$\underbrace{b_0 = \cdots = b_0}_{E_0 \text{ epochs}} \leq \underbrace{b_1 = \cdots = b_1}_{E_1 \text{ epochs}} \leq \cdots \leq \underbrace{b_M = \cdots = b_M = n}_{E_M \text{ epochs}}, \quad (11)$$

where $M \in \mathbb{N}$ and $E_i \in \mathbb{N}$ ($i \in [0:M]$). Accordingly, we have that the total number of steps for training is $T = \sum_{i \in [0:M]} \lceil \frac{n}{b_i} \rceil E_i$.

Theorem 2.1 leads us to the following theorem, the proof of which is given in Appendix B.2.

**Theorem 2.3 ($\epsilon$–approximation of GSAM with an increasing batch size and constant learning rate)** *Consider the sequence $(\boldsymbol{x}_t)$ generated by the mini-batch GSAM algorithm (Algorithm 1) with an increasing batch size $b_t \in (0, n]$ defined by (11) and a constant learning rate, $\eta_t = \eta \in (0, +\infty)$. Furthermore, let us assume that there exists a positive number $G$ such that $\max\{\sup_{t \in \mathbb{N} \cup \{0\}} \|\nabla f_S(\boldsymbol{x}_t + \hat{\boldsymbol{\epsilon}}_{S_t,\rho}(\boldsymbol{x}_t))\|_2, \sup_{t \in \mathbb{N} \cup \{0\}} \|\nabla \hat{f}_{S_t,\rho}^{\mathrm{SAM}}(\boldsymbol{x}_t)\|_2, \sup_{t \in \mathbb{N} \cup \{0\}} \|\nabla \hat{f}_{S,\rho}^{\mathrm{SAM}}(\boldsymbol{x}_t)\|_2, G_\perp\} \leq G$, where $G_\perp := \sup_{t \in \mathbb{N} \cup \{0\}} \|\nabla f_{S_t \perp}(\boldsymbol{x}_t)\|_2 < +\infty$ (see Theorem 2.1). Let $\epsilon > 0$ be the precision and let $b_0 > 0$, $\eta > 0$, $\alpha \in \mathbb{R}$, and $\rho \geq 0$ such that*

$$\eta \in \left[ \frac{12\sigma C}{\epsilon^2} \left( \frac{\rho G}{\sqrt{b_0}} + \frac{3\sigma}{nb_0} \sum_{i \in [n]} L_i \right), \frac{(|\alpha| + 1)^{-2} n^3 \epsilon^2}{6G^2 \sum_{i \in [n]} L_i \{ n^2 + 4C(\sum_{i \in [n]} L_i)^2 \}} \right], \quad (12)$$

$$\rho(|\alpha| + 1) \leq \frac{n\sqrt{b_0}\epsilon^2}{6G(CG\sqrt{b_0} + B\sigma) \sum_{i \in [n]} L_i}, \ \rho \leq \frac{nb_0 \epsilon^2}{2\sqrt{42}G\sqrt{n^2 + b_0^2} \sum_{i \in [n]} L_i}, \quad (13)$$

*where $B$ and $C$ are nonnegative constants. Then, there exists $t_0 \in \mathbb{N}$ such that, for all $T \geq t_0$,*

$$\min_{t \in [0:T-1]} \mathbb{E}\left[ \left\| \nabla \hat{f}_{S,\rho}^{\mathrm{SAM}}(\boldsymbol{x}_t) \right\|_2 \right] \leq \epsilon.$$

Theorem 2.3 indicates that the parameters $|\alpha|$ and $\rho$ in (13) become small and thereby achieve an $\epsilon$–approximation of GSAM. The setting of the small parameter $\rho$ is consistent with the definition of Problem 2.1 (see also (3)). Moreover, the setting also matches the numerical results in (Zhuang et al., 2022) that used small $|\alpha|$ and $\rho$. Using a small $\rho$ leads to the finding that $C$ and $B$ are approximately zero (see Propositions B.2 and B.3). In particular, $\rho = 0$ implies that $B = C = 0$. Hence, a constant learning rate $\eta$ satisfying (12) is approximately

$$\eta \in \left(0, \frac{n\epsilon^2}{6(|\alpha| + 1)^2 G^2 \sum_{i\in[n]} L_i}\right].$$

(14)

From (14), it would be appropriate to set a small $\eta$ in order to achieve an $\epsilon$-approximation of GSAM. In fact, the numerical results in (Zhuang et al., 2022) used small learning rates, such as $10^{-2}$, $10^{-3}$, and $10^{-5}$.

Since SGD (i.e., GSAM with $\alpha = \rho = 0$) satisfies (13), Theorem 2.3 guarantees that SGD is an $\epsilon$-approximation in the sense of $\min_{t\in[0:T-1]} \mathbb{E}[\|\nabla f_S(\boldsymbol{x}_t)\|_2] \leq \epsilon$. Moreover, using $\alpha = \rho = 0$ makes the upper bound of $\min_{t\in[0:T-1]} \mathbb{E}[\|\nabla f_S(\boldsymbol{x}_t)\|_2]$ $(= \min_{t\in[0:T-1]} \mathbb{E}[\|\nabla \hat{f}^{\text{SAM}}_{S,\rho}(\boldsymbol{x}_t)\|_2])$ smaller than using $\alpha \neq 0 \vee \rho \neq 0$. Hence, SGD using $\alpha = \rho = 0$ would minimize $\|\nabla f_S(\boldsymbol{x}_t)\|_2$ more quickly than would SAM/GSAM using $\alpha \neq 0 \vee \rho \neq 0$ (see also Figure 1 (Left) indicating that SGD minimizes $f_S$ more quickly than SAM/GSAM). Meanwhile, the previous results in (Foret et al., 2021; Zhuang et al., 2022) indicate that using $\alpha \neq 0 \vee \rho \neq 0$ leads to a better generalization than using $\alpha = \rho = 0$ (see Figure 1 (Right) and Table 2 indicating that SAM/GSAM with an increasing batch size has a higher generalization capability than SGD has with an increasing batch size).

### 2.4.2 CONSTANT BATCH SIZE AND DECAYING LEARNING RATE

Motivated by (Loshchilov & Hutter, 2017), we focus on a constant batch size defined for all $t \in \mathbb{N} \cup \{0\}$ by $b_t = b$ and examine a cosine-annealing rate scheduler defined by

$$\eta_t = \underline{\eta} + \frac{\overline{\eta} - \underline{\eta}}{2}\left(1 + \cos\left\lfloor \frac{t}{K} \right\rfloor \frac{\pi}{E}\right) \quad (t \in [0 : KE]),$$

(15)

where $\underline{\eta}$ and $\overline{\eta}$ are such that $0 \leq \underline{\eta} \leq \overline{\eta}$, $E$ is the number of epochs, and $K = \lceil \frac{n}{b} \rceil$ is the number of steps per epoch. We then have that the total number of steps for training is $T = KE$. The cosine-annealing learning rate (15) is updated per epoch and remains unchanged during $K$ steps.

Moreover, for a constant batch size $b_t = b$ $(t \in \mathbb{N} \cup \{0\})$, we examine a linear learning rate scheduler (Liu et al., 2020) defined by

$$\eta_t = \frac{\underline{\eta} - \overline{\eta}}{T}t + \overline{\eta} \quad (t \in [0 : T]),$$

(16)

where $\underline{\eta}$ and $\overline{\eta}$ are such that $0 \leq \underline{\eta} \leq \overline{\eta}$ and $T$ is the number of steps. The linear learning rate scheduler (16) is updated per step whose size decays linearly from step $0$ to $T$.

Theorem 2.1 leads us to the following theorem, the proof which is given in Appendix B.3 (The case where $\underline{\eta} > 0$ is also shown in Appendix B.3).

**Theorem 2.4 ($\epsilon$–approximation of GSAM with a constant batch size and decaying learning rate)** *Consider the sequence $(\boldsymbol{x}_t)$ generated by the mini-batch GSAM algorithm (Algorithm 1) with a constant batch size $b_t = b \in (0, n]$ and a decaying learning rate $\eta_t \in [\underline{\eta}, \overline{\eta}]$ defined by (15) or (16). Furthermore, let us assume that there exists a positive number $G$ defined as in Theorem 2.3. Let $\epsilon > 0$ be the precision and let $b > 0$, $\overline{\eta} > 0$ $(= \underline{\eta})$, $\alpha \in \mathbb{R}$, and $\rho \geq 0$ such that*

$$\overline{\eta} \in \begin{cases} \left[\frac{24\sigma C}{\epsilon^2}\left(\frac{\rho G}{\sqrt{b}} + \frac{3\sigma}{nb}\sum_{i\in[n]} L_i\right), \frac{2(|\alpha|+1)^{-2}n^3\epsilon^2}{9G^2\sum_{i\in[n]} L_i\{n^2+4C(\sum_{i\in[n]} L_i)^2\}}\right] & \text{if (15) is used,} \\ \left[\frac{24\sigma C}{\epsilon^2}\left(\frac{\rho G}{\sqrt{b}} + \frac{3\sigma}{nb}\sum_{i\in[n]} L_i\right), \frac{(|\alpha|+1)^{-2}n^3\epsilon^2}{4G^2\sum_{i\in[n]} L_i\{n^2+4C(\sum_{i\in[n]} L_i)^2\}}\right] & \text{if (16) is used,} \end{cases}$$

(17)

$$\rho(|\alpha| + 1) \leq \frac{n\sqrt{b}\epsilon^2}{6G(CG\sqrt{b} + B\sigma)\sum_{i\in[n]} L_i}, \; \rho \leq \frac{nb\epsilon^2}{2\sqrt{42}G\sqrt{n^2 + b^2}\sum_{i\in[n]} L_i},$$

(18)

*where $B$ and $C$ are nonnegative constants. Then, there exists $t_0 \in \mathbb{N}$ such that, for all $T \geq t_0$,*

$$\min_{t\in[0:T-1]} \mathbb{E}\left[\left\|\nabla \hat{f}^{\text{SAM}}_{S,\rho}(\boldsymbol{x}_t)\right\|_2\right] \leq \epsilon.$$

Theorem 2.4 indicates that the parameters $|\alpha|$ and $\rho$ in (18) become small and thereby achieve an $\epsilon$–approximation of GSAM, as also seen in Theorem 2.3. A discussion similar to the one showing (14) implies that the maximum learning rate $\overline{\eta}$ satisfying (17) using a small $\rho$ is approximately

$$\overline{\eta} \in \begin{cases} \left(0, \frac{2(|\alpha|+1)^{-2}n\epsilon^2}{9G^2 \sum_{i \in [n]} L_i}\right] & \text{if (15) is used,} \\ \left(0, \frac{(|\alpha|+1)^{-2}n\epsilon^2}{4G^2 \sum_{i \in [n]} L_i}\right] & \text{if (16) is used.} \end{cases} \tag{19}$$

From (19), it would be appropriate to set a small $\eta$ in order to achieve an $\epsilon$-approximation of GSAM. In fact, the numerical results in (Zhuang et al., 2022) used small values of $\overline{\eta}$, such as $1.6$ and $3 \times 10^{-3}$.

Theorem 2.4 guarantees that SGD is an $\epsilon$-approximation in the sense of $\min_{t \in [0:T-1]} \mathbb{E}[\|\nabla f_S(\boldsymbol{x}_t)\|_2] \leq \epsilon$. Moreover, using $\alpha = \rho = 0$ makes the upper bound of $\min_{t \in [0:T-1]} \mathbb{E}[\|\nabla f_S(\boldsymbol{x}_t)\|_2]$ smaller than when using $\alpha \neq 0 \vee \rho \neq 0$. Hence, SGD using $\alpha = \rho = 0$ would minimize $\|\nabla f_S(\boldsymbol{x}_t)\|_2$ more quickly than SAM/GSAM using $\alpha \neq 0 \vee \rho \neq 0$ (see also Figure 2 (Left) indicating that SGD minimizes $f_S$ more quickly than SAM/GSAM). Meanwhile, the previous results in (Foret et al., 2021; Zhuang et al., 2022) indicate that using $\alpha \neq 0 \vee \rho \neq 0$ leads to a higher generalization capability than using $\alpha = \rho = 0$ (see Table 2 which shows that the generalization capability of SAM/GSAM+C has a higher than that of SGD+C).

## 3 NUMERICAL RESULTS

We used a computer equipped with NVIDIA GeForce RTX 4090×2GPUs and an Intel Core i9 13900KF CPU. The software environment was Python 3.10.12, PyTorch 2.1.0, and CUDA 12.2. The solid lines in the figures represent the mean value and the shaded areas represent the maximum and minimum over three runs.

**Training Wide-ResNet28-10 on CIFAR100** We set $E = 200$, $\eta = \overline{\eta} = 0.1$, and $\underline{\eta} = 0.001$. We trained Wide-ResNet-28-10 on the CIFAR100 dataset (see Appendix C for an explanation of training ResNet-18 on the CIFAR100 dataset). The parameters, $\alpha = 0.02$ and $\rho = 0.05$, were determined by conducting a grid search of $\alpha \in \{0.01, 0.02, 0.03\}$ and $\rho \in \{0.01, 0.02, 0.03, 0.04, 0.05\}$. Figure 1 compares the use of an increasing batch size $[8, 16, 32, 64, 128]$ (SGD/SAM/GSAM + increasing_batch) with the use of a constant batch size $128$ (SGD/SAM/GSAM) for a fixed learning rate, $0.1$. SGD/SAM/GSAM + increasing_batch decreased the empirical loss (Figure 1 (Left)) and achieved higher test accuracies compared with SGD/SAM/GSAM (Figure 1 (Right)). Figure 2 compares the use of a cosine-annealing learning rate defined by (15) (SGD/SAM/GSAM + Cosine) with the use of a constant learning rate, $0.1$ (SGD/SAM/GSAM) for a fixed batch size $128$. SAM/GSAM + Cosine decreased the empirical loss (Figure 2 (Left)) and achieved higher test accuracies compared with SGD/SAM/GSAM (Figure 2 (Right)).

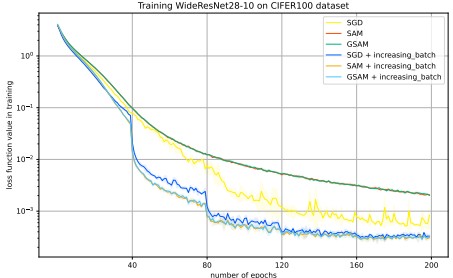
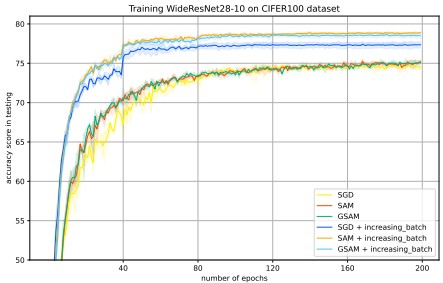

Figure 1: (Left) Loss function value in training and (Right) accuracy score in testing for the algorithms versus the number of epochs in training Wide-ResNet-28-10 on the CIFAR100 dataset. The learning rate of each algorithm was fixed at 0.1. In SGD/SAM/GSAM, the batch size was fixed at 128. In SGD/SAM/GSAM + increasing_batch, the batch size was set at 8 for the first 40 epochs and then it was doubled every 40 epochs afterwards, i.e., to 16 for epochs 41-80, 32 for epochs 81-120, etc.

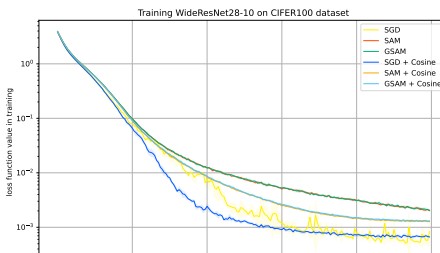 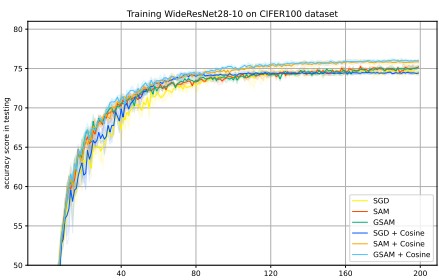

Figure 2: (Left) Loss function value in training and (Right) accuracy score in testing for the algorithms versus the number of epochs in training Wide-ResNet28-10 on the CIFAR100 dataset. The batch size of each algorithm was fixed at 128. In SGD/SAM/GSAM, the constant learning rate was fixed at 0.1. In SGD/SAM/GSAM + Cosine, the maximum learning rate was 0.1 and the minimum learning rate was 0.001.

Table 2: Mean values of the test errors (Test Error) and the worst-case $\ell_\infty$ adaptive sharpness (Sharpness) for the parameter obtained by the algorithms at 200 epochs in training Wide-ResNet28-10 on the CIFAR100 dataset. "(algorithm)+B" refers to "(algorithm) + increasing_batch" used in Figure 1, and "(algorithm)+C" refers to "(algorithm) + Cosine" used in Figure 2.

|  | SGD | SAM | GSAM | SGD+B | SAM+B | GSAM+B | SGD+C | SAM+C | GSAM+C |
|---|---|---|---|---|---|---|---|---|---|
| Test Error | 25.62 | 24.78 | 24.94 | 22.65 | **21.10** | 21.50 | 25.57 | 24.16 | 24.00 |
| Sharpness | 1113.26 | 456.20 | 435.17 | 22.72 | **10.99** | 12.37 | 1148.09 | 687.44 | 665.13 |

Table 2 summarizes the mean values of the test errors and the worst-case $\ell_\infty$ adaptive sharpness defined by (Andriushchenko et al., 2023b, (1)) for the parameters $c = (1, 1, \cdots, 1)^\top$ and $\rho = 0.0002$ obtained by the algorithm after 200 epochs. SAM+B (SAM + increasing_batch) had the highest test accuracy and the lowest sharpness, which implies that SAM+B approximated a flatter local minimum. The table indicates that increasing batch sizes could avoid sharp local minima to which the algorithms using the constant and cosine-annealing learning rates converged.

**Training ViT-Tiny on CIFAR100** We set $E = 100$ and a learning rate of $\overline{\eta} = 0.001$ with an initial learning rate of $0.00001$ and linear warmup during 10 epochs. We trained ViT-Tiny on the CIFAR100 dataset (see Appendix D for the ViT-Tiny model). We used Adam (Kingma & Ba, 2015) with $\beta_1 = 0.9$, $\beta_2 = 0.999$ and a weight decay of 0.05 as the base algorithm. The parameters, $\alpha = 0.1$ and $\rho = 0.6$, were determined by conducting a grid search of $\alpha \in \{0.1, 0.2, 0.3\}$ and $\rho \in \{0.1, 0.2, 0.3, 0.4, 0.5, 0.6\}$. We used the data extension and regularization technique in (Lee et al., 2021). Figure 3 compares the use of an increasing batch size $[64, 128, 256, 512]$ (Adam/SAM/GSAM + increasing_batch) with the use of a constant batch size 128 (Adam/SAM/GSAM) for a fixed learning rate, 0.001. SAM + increasing_batch achieved higher test accuracies compared with Adam/SAM/GSAM (Figure 3 (Right)). Figure 4 compares the use of a cosine-annealing learning rate defined by (15) (Adam/SAM/GSAM + Cosine) with the use of a constant learning rate, 0.001, (Adam/SAM/GSAM) for a fixed batch size, 128. Adam + Cosine achieved higher test accuracies than Adam/SAM/GSAM (Figure 4 (Right)).

Table 3: Mean values of the test errors (Test Error) and the worst-case $\ell_\infty$ adaptive sharpness (Sharpness) for the parameter obtained by the algorithms at 100 epochs in training ViT-Tiny on the CIFAR100 dataset. "(algorithm)+B" refers to "(algorithm) + increasing_batch" in Figure 3, and "(algorithm)+C" refers to "(algorithm) + Cosine" in Figure 4.

|  | Adam | SAM | GSAM | Adam+B | SAM+B | GSAM+B | Adam+C | SAM+C | GSAM+C |
|---|---|---|---|---|---|---|---|---|---|
| Test Error | 31.62 | 29.20 | 29.81 | 29.26 | 28.45 | 29.10 | **27.06** | 28.18 | 28.90 |
| Sharpness | 0.28 | 0.16 | **0.15** | 0.24 | **0.15** | 0.16 | 0.42 | 0.17 | 0.17 |

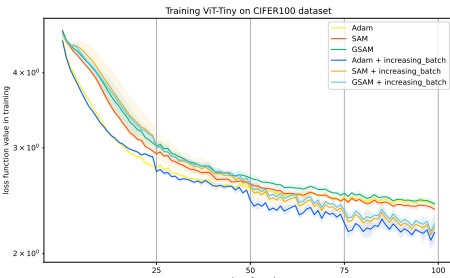 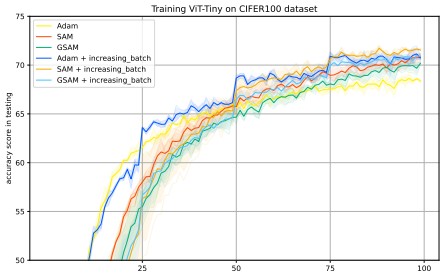

Figure 3: (Left) Loss function value in training and (Right) accuracy score in testing for the algorithms versus the number of epochs in training ViT-Tiny on the CIFAR100 dataset. The learning rate of each algorithm was fixed at 0.001 with an initial learning rate 0.00001 and linear warmup during 10 epochs. In Adam/SAM/GSAM, the batch size was fixed at 128. In Adam/SAM/GSAM + increasing batch, the batch size was set at 64 for the first 25 epochs and then it was doubled every 25 epochs afterwards, i.e., to 128 for epochs 26-50, 256 for epochs 51-75, etc.

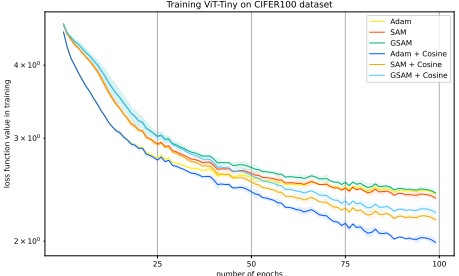 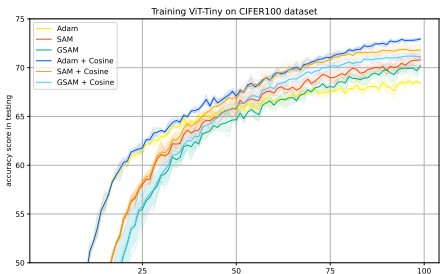

Figure 4: (Left) Loss function value in training and (Right) accuracy score in testing for the algorithms versus the number of epochs in training ViT-Tiny on the CIFAR100 dataset. The batch size of each algorithm was fixed at 128. In Adam/SAM/GSAM, the constant learning rate was fixed at 0.001 with an initial learning rate 0.00001 and linear warmup during the first 10 epochs. In Adam/SAM/GSAM + Cosine, the maximum learning rate was 0.001 and the minimum learning rate was 0.00001 with linear warmup during the first 10 epochs.

Table 3 summarizes the mean values of the test errors and the worst-case $\ell_\infty$ adaptive sharpness defined by (Andriushchenko et al., 2023b, (1)) for the parameters $c = (1, 1, \cdots, 1)^\top$ and $\rho = 0.0002$ obtained by the algorithm after $100$ epochs. The table indicates that SAM+B could avoid local minima to which the algorithms using the cosine-annealing learning rate converged.

## 4 CONCLUSION

First we gave upper and lower bounds of the search direction noise of the GSAM algorithm for solving the SAM problem. Then, we examined the GSAM algorithm with two mini-batch and learning rate schedulers based on the bounds: an increasing batch size and constant learning rate scheduler and a constant batch size and decaying learning rate scheduler. We performed convergence analyses on GSAM for the two schedulers. We also provided numerical results to support the analyses. The numerical results showed that, compared with SGD/Adam, SAM/GSAM with an increasing batch size and a constant learning rate converges to flatter local minima of the empirical loss functions for ResNets and ViT-Tiny on the CIFAR100 dataset.

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

## A PROOFS OF THEOREMS 2.1 AND 2.2

### A.1 PROPOSITIONS

We first give an upper bound of the variance of the stochastic gradient $\nabla f_{S_t}(\boldsymbol{x})$.

**Proposition A.1** *Under Assumption 2.1, we have that, for all $\boldsymbol{x} \in \mathbb{R}^d$ and all $t \in \mathbb{N} \cup \{0\}$,*

$$\mathbb{E}_{\boldsymbol{\xi}_t} \left[ \nabla f_{S_t}(\boldsymbol{x}) \Big| \hat{\boldsymbol{\xi}}_{t-1} \right] = \nabla f_S(\boldsymbol{x}),$$

$$\mathbb{V}_{\boldsymbol{\xi}_t} \left[ \nabla f_{S_t}(\boldsymbol{x}) \Big| \hat{\boldsymbol{\xi}}_{t-1} \right] = \mathbb{E}_{\boldsymbol{\xi}_t} \left[ \|\nabla f_{S_t}(\boldsymbol{x}) - \nabla f_S(\boldsymbol{x})\|_2^2 \Big| \hat{\boldsymbol{\xi}}_{t-1} \right] \leq \frac{\sigma^2}{b_t},$$

*where $\mathbb{E}_{\boldsymbol{\xi}_t}[\cdot|\hat{\boldsymbol{\xi}}_{t-1}]$ stands for the expectation with respect to $\boldsymbol{\xi}_t$ conditioned on $\boldsymbol{\xi}_{t-1} = \hat{\boldsymbol{\xi}}_{t-1}$.*

*Proof:* Let $\boldsymbol{x} \in \mathbb{R}^d$ and $t \in \mathbb{N} \cup \{0\}$. Assumption 2.1(A3) ensures that

$$\mathbb{E}_{\boldsymbol{\xi}_t} \left[ \nabla f_{S_t}(\boldsymbol{x}) \Big| \hat{\boldsymbol{\xi}}_{t-1} \right] = \mathbb{E}_{\boldsymbol{\xi}_t} \left[ \frac{1}{b_t} \sum_{i \in [b_t]} \nabla f_{\xi_{t,i}}(\boldsymbol{x}) \Bigg| \hat{\boldsymbol{\xi}}_{t-1} \right] = \frac{1}{b_t} \sum_{i \in [b_t]} \mathbb{E}_{\xi_{t,i}} \left[ \nabla f_{\xi_{t,i}}(\boldsymbol{x}) \Big| \hat{\boldsymbol{\xi}}_{t-1} \right],$$

which, together with Assumption 2.1(A2)(i) and the independence of $\boldsymbol{\xi}_t$ and $\boldsymbol{\xi}_{t-1}$, implies that

$$\mathbb{E}_{\boldsymbol{\xi}_t} \left[ \nabla f_{S_t}(\boldsymbol{x}) \Big| \hat{\boldsymbol{\xi}}_{t-1} \right] = \nabla f_S(\boldsymbol{x}).$$

Assumption 2.1(A3) implies that

$$\mathbb{V}_{\boldsymbol{\xi}_t} \left[ \nabla f_{S_t}(\boldsymbol{x}) \Big| \hat{\boldsymbol{\xi}}_{t-1} \right] = \mathbb{E}_{\boldsymbol{\xi}_t} \left[ \|\nabla f_{S_t}(\boldsymbol{x}) - \nabla f_S(\boldsymbol{x})\|_2^2 \Big| \hat{\boldsymbol{\xi}}_{t-1} \right]$$

$$= \mathbb{E}_{\boldsymbol{\xi}_t} \left[ \left\| \frac{1}{b_t} \sum_{i \in [b_t]} \nabla f_{\xi_{t,i}}(\boldsymbol{x}) - \nabla f_S(\boldsymbol{x}) \right\|_2^2 \Bigg| \hat{\boldsymbol{\xi}}_{t-1} \right]$$

$$= \frac{1}{b_t^2} \mathbb{E}_{\boldsymbol{\xi}_t} \left[ \left\| \sum_{i \in [b_t]} \left( \nabla f_{\xi_{t,i}}(\boldsymbol{x}) - \nabla f_S(\boldsymbol{x}) \right) \right\|_2^2 \Bigg| \hat{\boldsymbol{\xi}}_{t-1} \right].$$

From the independence of $\xi_{t,i}$ and $\xi_{t,j}$ $(i \neq j)$, for all $i, j \in [b_t]$ with $i \neq j$,

$$\mathbb{E}_{\xi_{t,i}}[\langle \nabla f_{\xi_{t,i}}(\boldsymbol{x}) - \nabla f_S(\boldsymbol{x}), \nabla f_{\xi_{t,j}}(\boldsymbol{x}) - \nabla f_S(\boldsymbol{x}) \rangle_2 | \hat{\boldsymbol{\xi}}_{t-1}]$$

$$= \langle \mathbb{E}_{\xi_{t,i}}[\nabla f_{\xi_{t,i}}(\boldsymbol{x}) | \hat{\boldsymbol{\xi}}_{t-1}] - \mathbb{E}_{\xi_{t,i}}[\nabla f_S(\boldsymbol{x}) | \hat{\boldsymbol{\xi}}_{t-1}], \nabla f_{\xi_{t,j}}(\boldsymbol{x}) - \nabla f_S(\boldsymbol{x}) \rangle_2 = 0.$$

Hence, Assumption 2.1(A2)(ii) guarantees that

$$\mathbb{V}_{\boldsymbol{\xi}_t} \left[ \nabla f_{S_t}(\boldsymbol{x}) \Big| \hat{\boldsymbol{\xi}}_{t-1} \right] = \frac{1}{b_t^2} \sum_{i \in [b_t]} \mathbb{E}_{\xi_{t,i}} \left[ \|\nabla f_{\xi_{t,i}}(\boldsymbol{x}) - \nabla f_S(\boldsymbol{x})\|_2^2 \Big| \hat{\boldsymbol{\xi}}_{t-1} \right]$$

$$\leq \frac{\sigma^2 b_t}{b_t^2} = \frac{\sigma^2}{b_t},$$

which completes the proof. □

We will use the following proposition to prove Theorem 2.1.

**Proposition A.2** *(Ortega & Rheinboldt, 2000, 3.2.6, (10)) Let $f : \mathbb{R}^d \to \mathbb{R}$ be twice differentiable. Then, for all $\boldsymbol{x}, \boldsymbol{y} \in \mathbb{R}^d$,*

$$\nabla f(\boldsymbol{y}) = \nabla f(\boldsymbol{x}) + \int_0^1 \nabla^2 f(\boldsymbol{x} + t(\boldsymbol{y} - \boldsymbol{x}))(\boldsymbol{y} - \boldsymbol{x}) \mathrm{d}t.$$

A.2   PROOF OF THEOREM 2.1

We will use Propositions A.1 and A.2 to prove Theorem 2.1.

Let $t \in \mathbb{N} \cup \{0\}$ and $b < n$ and suppose that $\boldsymbol{x}_t$ generated by Algorithm 1 satisfies $\nabla f_{S_t}(\boldsymbol{x}_t) \neq \boldsymbol{0}$ and $\nabla f_S(\boldsymbol{x}_t) \neq \boldsymbol{0}$. Then, we have

$$
\begin{aligned}
\|\hat{\boldsymbol{\omega}}_t\|_2^2 &= \left\| \nabla \hat{f}_{S_t,\rho}^{\mathrm{SAM}}(\boldsymbol{x}_t) - \nabla \hat{f}_{S,\rho}^{\mathrm{SAM}}(\boldsymbol{x}_t) \right\|_2^2 \\
&\overset{=}{\scriptstyle(7)} \left\| \nabla f_{S_t}\left( \boldsymbol{x}_t + \rho \frac{\nabla f_{S_t}(\boldsymbol{x}_t)}{\|\nabla f_{S_t}(\boldsymbol{x}_t)\|_2} \right) - \nabla f_S\left( \boldsymbol{x}_t + \rho \frac{\nabla f_S(\boldsymbol{x}_t)}{\|\nabla f_S(\boldsymbol{x}_t)\|_2} \right) \right\|_2^2 \\
&= \left\| \nabla f_{S_t}(\boldsymbol{x}_t) + \int_0^1 \nabla^2 f_{S_t}\left( \boldsymbol{x}_t + \rho s \frac{\nabla f_{S_t}(\boldsymbol{x}_t)}{\|\nabla f_{S_t}(\boldsymbol{x}_t)\|_2} \right) \rho \frac{\nabla f_{S_t}(\boldsymbol{x}_t)}{\|\nabla f_{S_t}(\boldsymbol{x}_t)\|_2} \mathrm{d}s \right. \\
&\quad \left. - \left( \nabla f_S(\boldsymbol{x}_t) + \int_0^1 \nabla^2 f_S\left( \boldsymbol{x}_t + \rho s \frac{\nabla f_S(\boldsymbol{x}_t)}{\|\nabla f_S(\boldsymbol{x}_t)\|_2} \right) \rho \frac{\nabla f_S(\boldsymbol{x}_t)}{\|\nabla f_S(\boldsymbol{x}_t)\|_2} \mathrm{d}s \right) \right\|_2^2,
\end{aligned}
\tag{20}
$$

where the third equation comes from Proposition A.2. From $\|\boldsymbol{x} + \boldsymbol{y}\|_2^2 \leq 2\|\boldsymbol{x}\|_2^2 + 2\|\boldsymbol{y}\|_2^2$ $(\boldsymbol{x}, \boldsymbol{y} \in \mathbb{R}^d)$, we have

$$
\begin{aligned}
\|\hat{\boldsymbol{\omega}}_t\|_2^2 &\leq 2\|\nabla f_{S_t}(\boldsymbol{x}_t) - \nabla f_S(\boldsymbol{x}_t)\|_2^2 \\
&\quad + 4\left\| \int_0^1 \nabla^2 f_{S_t}\left( \boldsymbol{x}_t + \rho s \frac{\nabla f_{S_t}(\boldsymbol{x}_t)}{\|\nabla f_{S_t}(\boldsymbol{x}_t)\|_2} \right) \rho \frac{\nabla f_{S_t}(\boldsymbol{x}_t)}{\|\nabla f_{S_t}(\boldsymbol{x}_t)\|_2} \mathrm{d}s \right\|_2^2 \\
&\quad + 4\left\| \int_0^1 \nabla^2 f_S\left( \boldsymbol{x}_t + \rho s \frac{\nabla f_S(\boldsymbol{x}_t)}{\|\nabla f_S(\boldsymbol{x}_t)\|_2} \right) \rho \frac{\nabla f_S(\boldsymbol{x}_t)}{\|\nabla f_S(\boldsymbol{x}_t)\|_2} \mathrm{d}s \right\|_2^2,
\end{aligned}
$$

which, together with the property of $\|\cdot\|_2$, implies that

$$
\begin{aligned}
\|\hat{\boldsymbol{\omega}}_t\|_2^2 &\leq 2\|\nabla f_{S_t}(\boldsymbol{x}_t) - \nabla f_S(\boldsymbol{x}_t)\|_2^2 \\
&\quad + 4\left( \rho \int_0^1 \left\| \nabla^2 f_{S_t}\left( \boldsymbol{x}_t + \rho s \frac{\nabla f_{S_t}(\boldsymbol{x}_t)}{\|\nabla f_{S_t}(\boldsymbol{x}_t)\|_2} \right) \right\|_2 \mathrm{d}s \right)^2 \\
&\quad + 4\left( \rho \int_0^1 \left\| \nabla^2 f_S\left( \boldsymbol{x}_t + \rho s \frac{\nabla f_S(\boldsymbol{x}_t)}{\|\nabla f_S(\boldsymbol{x}_t)\|_2} \right) \right\|_2 \mathrm{d}s \right)^2.
\end{aligned}
\tag{21}
$$

Meanwhile, the triangle inequality and the $L_i$–smoothness of $f_i$ (see (A1)) ensure that, for all $\boldsymbol{x}, \boldsymbol{y} \in \mathbb{R}^d$,

$$
\begin{aligned}
\|\nabla f_{S_t}(\boldsymbol{x}) - \nabla f_{S_t}(\boldsymbol{y})\|_2 &= \left\| \frac{1}{b_t} \sum_{i \in [b_t]} (\nabla f_{\xi_{t,i}}(\boldsymbol{x}) - \nabla f_{\xi_{t,i}}(\boldsymbol{y})) \right\|_2 \leq \frac{1}{b_t} \sum_{i \in [b_t]} \left\| \nabla f_{\xi_{t,i}}(\boldsymbol{x}) - \nabla f_{\xi_{t,i}}(\boldsymbol{y}) \right\|_2 \\
&\leq \frac{1}{b_t} \sum_{i \in [b_t]} L_{\xi_{t,i}} \|\boldsymbol{x} - \boldsymbol{y}\|_2 \leq \frac{1}{b_t} \sum_{i \in [n]} L_i \|\boldsymbol{x} - \boldsymbol{y}\|_2,
\end{aligned}
$$

which implies that, for all $\boldsymbol{x} \in \mathbb{R}^d$, $\|\nabla^2 f_{S_t}(\boldsymbol{x})\|_2 \leq b_t^{-1} \sum_{i \in [n]} L_i$. A discussion similar to the one showing that $\nabla f_{S_t}$ is $b_t^{-1} \sum_{i \in [n]} L_i$–smooth ensures that $\nabla f_S$ is $n^{-1} \sum_{i \in [n]} L_i$–smooth, which in turn implies that, for all $\boldsymbol{x} \in \mathbb{R}^d$, $\|\nabla^2 f_S(\boldsymbol{x})\|_2 \leq n^{-1} \sum_{i \in [n]} L_i$. Accordingly, (21) guarantees that

$$
\|\hat{\boldsymbol{\omega}}_t\|_2^2 \leq 2\|\nabla f_{S_t}(\boldsymbol{x}_t) - \nabla f_S(\boldsymbol{x}_t)\|_2^2 + \frac{4\rho^2}{b_t^2}\left( \sum_{i \in [n]} L_i \right)^2 + \frac{4\rho^2}{n^2}\left( \sum_{i \in [n]} L_i \right)^2.
\tag{22}
$$

Taking the expectation with respect to $\boldsymbol{\xi}_t$ conditioned on $\boldsymbol{\xi}_{t-1} = \hat{\boldsymbol{\xi}}_{t-1}$ on both sides of (22) ensures that

$$
\mathbb{E}_{\boldsymbol{\xi}_t}[\|\hat{\boldsymbol{\omega}}_t\|_2^2 | \hat{\boldsymbol{\xi}}_{t-1}] \leq 2\mathbb{E}_{\boldsymbol{\xi}_t}[\|\nabla f_{S_t}(\boldsymbol{x}_t) - \nabla f_S(\boldsymbol{x}_t)\|_2^2 | \hat{\boldsymbol{\xi}}_{t-1}] + \frac{4\rho^2}{b_t^2}\left( \sum_{i \in [n]} L_i \right)^2 + \frac{4\rho^2}{n^2}\left( \sum_{i \in [n]} L_i \right)^2,
$$

which, together with Proposition A.1, implies that

$$
\mathbb{E}_{\boldsymbol{\xi}_t}[\|\hat{\boldsymbol{\omega}}_t\|_2^2 | \hat{\boldsymbol{\xi}}_{t-1}] \leq \frac{2\sigma^2}{b_t} + \frac{4\rho^2}{b_t^2}\left( \sum_{i \in [n]} L_i \right)^2 + \frac{4\rho^2}{n^2}\left( \sum_{i \in [n]} L_i \right)^2.
$$

Since $\boldsymbol{\xi}_t$ is independent of $\boldsymbol{\xi}_{t-1}$, we have

$$\mathbb{E}_{\boldsymbol{\xi}_{t-1}}\mathbb{E}_{\boldsymbol{\xi}_t}[\|\hat{\boldsymbol{\omega}}_t\|_2^2] = \mathbb{E}_{\boldsymbol{\xi}_{t-1}}[\mathbb{E}_{\boldsymbol{\xi}_t}[\|\hat{\boldsymbol{\omega}}_t\|_2^2|\boldsymbol{\xi}_{t-1}]] \leq \frac{2\sigma^2}{b_t} + \frac{4\rho^2}{b_t^2}\left(\sum_{i\in[n]}L_i\right)^2 + \frac{4\rho^2}{n^2}\left(\sum_{i\in[n]}L_i\right)^2,$$

which, together with $\mathbb{E} = \mathbb{E}_{\boldsymbol{\xi}_0}\mathbb{E}_{\boldsymbol{\xi}_1}\cdots\mathbb{E}_{\boldsymbol{\xi}_t}$, implies that

$$\mathbb{E}[\|\hat{\boldsymbol{\omega}}_t\|_2^2] \leq \frac{2\sigma^2}{b_t} + \frac{4\rho^2}{b_t^2}\left(\sum_{i\in[n]}L_i\right)^2 + \frac{4\rho^2}{n^2}\left(\sum_{i\in[n]}L_i\right)^2. \tag{23}$$

Suppose that $\boldsymbol{x}_t$ generated by Algorithm 1 satisfies either $\nabla f_{S_t}(\boldsymbol{x}_t) = \mathbf{0}$ or $\nabla f_S(\boldsymbol{x}_t) = \mathbf{0}$. Let $\nabla f_{S_t}(\boldsymbol{x}_t) = \mathbf{0}$. A discussion similar to the one obtaining (20) and (21), together with (7), ensures that

$$\|\hat{\boldsymbol{\omega}}_t\|_2^2 = \left\|\nabla\hat{f}_{S_t,\rho}^{\text{SAM}}(\boldsymbol{x}_t) - \nabla\hat{f}_{S,\rho}^{\text{SAM}}(\boldsymbol{x}_t)\right\|_2^2$$

$$= \left\|\nabla f_{S_t}(\boldsymbol{x}_t + \boldsymbol{u}) - \nabla f_S\left(\boldsymbol{x}_t + \rho\frac{\nabla f_S(\boldsymbol{x}_t)}{\|\nabla f_S(\boldsymbol{x}_t)\|_2}\right)\right\|_2^2$$

$$= \left\|\nabla f_{S_t}(\boldsymbol{x}_t) + \int_0^1 \nabla^2 f_{S_t}(\boldsymbol{x}_t + s\boldsymbol{u})\boldsymbol{u}\,\mathrm{d}s\right.$$

$$\left. - \left(\nabla f_S(\boldsymbol{x}_t) + \int_0^1 \nabla^2 f_S\left(\boldsymbol{x}_t + \rho s\frac{\nabla f_S(\boldsymbol{x}_t)}{\|\nabla f_S(\boldsymbol{x}_t)\|_2}\right)\rho\frac{\nabla f_S(\boldsymbol{x}_t)}{\|\nabla f_S(\boldsymbol{x}_t)\|_2}\,\mathrm{d}s\right)\right\|_2^2,$$

which, together with $\|\boldsymbol{u}\|_2 \leq \rho$, implies that

$$\|\hat{\boldsymbol{\omega}}_t\|_2^2 \leq 2\|\nabla f_{S_t}(\boldsymbol{x}_t) - \nabla f_S(\boldsymbol{x}_t)\|_2^2$$

$$+ 4\left(\rho\int_0^1 \left\|\nabla^2 f_{S_t}(\boldsymbol{x}_t + s\boldsymbol{u})\right\|_2\,\mathrm{d}s\right)^2$$

$$+ 4\left(\rho\int_0^1 \left\|\nabla^2 f_S\left(\boldsymbol{x}_t + \rho s\frac{\nabla f_S(\boldsymbol{x}_t)}{\|\nabla f_S(\boldsymbol{x}_t)\|_2}\right)\right\|_2\,\mathrm{d}s\right)^2.$$

Hence, the same discussion as in (22) leads to the finding that

$$\|\hat{\boldsymbol{\omega}}_t\|_2^2 \leq 2\|\nabla f_{S_t}(\boldsymbol{x}_t) - \nabla f_S(\boldsymbol{x}_t)\|_2^2 + \frac{4\rho^2}{b_t^2}\left(\sum_{i\in[n]}L_i\right)^2 + \frac{4\rho^2}{n^2}\left(\sum_{i\in[n]}L_i\right)^2.$$

Accordingly, Proposition A.1 and a discussion similar to the one showing (23) imply that (23) holds in the case of $\nabla f_{S_t}(\boldsymbol{x}_t) = \mathbf{0}$. Moreover, it ensures that (23) holds in the case of $\nabla f_S(\boldsymbol{x}_t) = \mathbf{0}$. Therefore, we have

$$\mathbb{E}[\|\hat{\boldsymbol{\omega}}_t\|_2] \leq \sqrt{\frac{2\sigma^2}{b_t} + \frac{4\rho^2}{b_t^2}\left(\sum_{i\in[n]}L_i\right)^2 + \frac{4\rho^2}{n^2}\left(\sum_{i\in[n]}L_i\right)^2}. \tag{24}$$

We reach the desired result for when $b_t < n$ in Theorem 2.1 from $\|\boldsymbol{\omega}_t\|_2 \leq \|\hat{\boldsymbol{\omega}}_t\|_2 + |\alpha|G_\perp$ and (24). We reach the desired result for when $b_t = n$ from $\|\hat{\boldsymbol{\omega}}_t\|_2^2 = 0$. This completes the proof. $\square$

### A.3 Proof of Theorem 2.2

Let $t \in \mathbb{N} \cup \{0\}$ and $b < n$ and suppose that $\boldsymbol{x}_t$ generated by Algorithm 1 satisfies $\nabla f_{S_t}(\boldsymbol{x}_t) \neq \mathbf{0}$ and $\nabla f_S(\boldsymbol{x}_t) \neq \mathbf{0}$. From $|\alpha|\|\nabla f_{S_t\perp}(\boldsymbol{x}_t)\|_2 \leq \|\hat{\boldsymbol{\omega}}_t\|_2$, we have

$$\|\boldsymbol{\omega}_t\|_2 \geq \|\hat{\boldsymbol{\omega}}_t\|_2 - |\alpha|\|\nabla f_{S_t\perp}(\boldsymbol{x}_t)\|_2 \geq \|\hat{\boldsymbol{\omega}}_t\|_2 - |\alpha|G_\perp.$$

From (20), we have

$$\|\hat{\boldsymbol{\omega}}_t\|_2 = \left\|\nabla\hat{f}_{S_t,\rho}^{\text{SAM}}(\boldsymbol{x}_t) - \nabla\hat{f}_{S,\rho}^{\text{SAM}}(\boldsymbol{x}_t)\right\|_2$$

$$
\geq \left| \|\nabla f_{S_t}(\boldsymbol{x}_t) - \nabla f_S(\boldsymbol{x}_t)\|_2 - \left\| \int_0^1 \nabla^2 f_{S_t}\left(\boldsymbol{x}_t + \rho s \frac{\nabla f_{S_t}(\boldsymbol{x}_t)}{\|\nabla f_{S_t}(\boldsymbol{x}_t)\|_2}\right) \rho \frac{\nabla f_{S_t}(\boldsymbol{x}_t)}{\|\nabla f_{S_t}(\boldsymbol{x}_t)\|_2} \mathrm{d}s \right. \right.
$$

$$
\left. \left. - \int_0^1 \nabla^2 f_S\left(\boldsymbol{x}_t + \rho s \frac{\nabla f_S(\boldsymbol{x}_t)}{\|\nabla f_S(\boldsymbol{x}_t)\|_2}\right) \rho \frac{\nabla f_S(\boldsymbol{x}_t)}{\|\nabla f_S(\boldsymbol{x}_t)\|_2} \mathrm{d}s \right\|_2 \right| =: |A_t|. \tag{25}
$$

When $A_t \geq 0$,

$$
\|\hat{\boldsymbol{\omega}}_t\|_2 \geq \|\nabla f_{S_t}(\boldsymbol{x}_t) - \nabla f_S(\boldsymbol{x}_t)\|_2 - \left\| \int_0^1 \nabla^2 f_{S_t}\left(\boldsymbol{x}_t + \rho s \frac{\nabla f_{S_t}(\boldsymbol{x}_t)}{\|\nabla f_{S_t}(\boldsymbol{x}_t)\|_2}\right) \rho \frac{\nabla f_{S_t}(\boldsymbol{x}_t)}{\|\nabla f_{S_t}(\boldsymbol{x}_t)\|_2} \mathrm{d}s \right.
$$

$$
\left. - \int_0^1 \nabla^2 f_S\left(\boldsymbol{x}_t + \rho s \frac{\nabla f_S(\boldsymbol{x}_t)}{\|\nabla f_S(\boldsymbol{x}_t)\|_2}\right) \rho \frac{\nabla f_S(\boldsymbol{x}_t)}{\|\nabla f_S(\boldsymbol{x}_t)\|_2} \mathrm{d}s \right\|_2
$$

$$
\geq \|\nabla f_{S_t}(\boldsymbol{x}_t) - \nabla f_S(\boldsymbol{x}_t)\|_2 - \rho \left(\frac{1}{b_t} + \frac{1}{n}\right) \sum_{i \in [n]} L_i,
$$

where the second inequality comes from (21) and (22). A similar discussion to the one in (23), together with Proposition A.1, implies that there exists $c_t \in [0, 1]$ such that

$$
\mathbb{E}[\|\hat{\boldsymbol{\omega}}_t\|_2] \geq \frac{c_t \sigma}{\sqrt{b_t}} - \rho \left(\frac{1}{b_t} + \frac{1}{n}\right) \sum_{i \in [n]} L_i.
$$

Accordingly, we have

$$
\mathbb{E}[\|\boldsymbol{\omega}_t\|_2] \geq \frac{c_t \sigma}{\sqrt{b_t}} - \rho \left(\frac{1}{b_t} + \frac{1}{n}\right) \sum_{i \in [n]} L_i - |\alpha| G_\perp. \tag{26}
$$

Furthermore, when $A_t < 0$, we have

$$
\|\hat{\boldsymbol{\omega}}_t\|_2 \geq \left\| \int_0^1 \nabla^2 f_{S_t}\left(\boldsymbol{x}_t + \rho s \frac{\nabla f_{S_t}(\boldsymbol{x}_t)}{\|\nabla f_{S_t}(\boldsymbol{x}_t)\|_2}\right) \rho \frac{\nabla f_{S_t}(\boldsymbol{x}_t)}{\|\nabla f_{S_t}(\boldsymbol{x}_t)\|_2} \mathrm{d}s \right.
$$

$$
\left. - \int_0^1 \nabla^2 f_S\left(\boldsymbol{x}_t + \rho s \frac{\nabla f_S(\boldsymbol{x}_t)}{\|\nabla f_S(\boldsymbol{x}_t)\|_2}\right) \rho \frac{\nabla f_S(\boldsymbol{x}_t)}{\|\nabla f_S(\boldsymbol{x}_t)\|_2} \mathrm{d}s \right\|_2 - \|\nabla f_{S_t}(\boldsymbol{x}_t) - \nabla f_S(\boldsymbol{x}_t)\|_2
$$

$$
\geq \left| \left\| \int_0^1 \nabla^2 f_{S_t}\left(\boldsymbol{x}_t + \rho s \frac{\nabla f_{S_t}(\boldsymbol{x}_t)}{\|\nabla f_{S_t}(\boldsymbol{x}_t)\|_2}\right) \rho \frac{\nabla f_{S_t}(\boldsymbol{x}_t)}{\|\nabla f_{S_t}(\boldsymbol{x}_t)\|_2} \mathrm{d}s \right\|_2 \right.
$$

$$
\left. - \left\| \int_0^1 \nabla^2 f_S\left(\boldsymbol{x}_t + \rho s \frac{\nabla f_S(\boldsymbol{x}_t)}{\|\nabla f_S(\boldsymbol{x}_t)\|_2}\right) \rho \frac{\nabla f_S(\boldsymbol{x}_t)}{\|\nabla f_S(\boldsymbol{x}_t)\|_2} \mathrm{d}s \right\|_2 \right| - \|\nabla f_{S_t}(\boldsymbol{x}_t) - \nabla f_S(\boldsymbol{x}_t)\|_2,
$$

which, together with (21) and (22), implies that there exists $d_t \in (0, 1]$ such that

$$
\|\hat{\boldsymbol{\omega}}_t\|_2 \geq \rho \left(\frac{d_t}{b_t} - \frac{1}{n}\right) \sum_{i \in [n]} L_i - \|\nabla f_{S_t}(\boldsymbol{x}_t) - \nabla f_S(\boldsymbol{x}_t)\|_2.
$$

A similar discussion to the one in (23), together with Proposition A.1, implies that

$$
\mathbb{E}[\|\hat{\boldsymbol{\omega}}_t\|_2] \geq \rho \left(\frac{d_t}{b_t} - \frac{1}{n}\right) \sum_{i \in [n]} L_i - \frac{\sigma}{\sqrt{b_t}}.
$$

Hence,

$$
\mathbb{E}[\|\boldsymbol{\omega}_t\|_2] \geq \rho \left(\frac{d_t}{b_t} - \frac{1}{n}\right) \sum_{i \in [n]} L_i - \frac{\sigma}{\sqrt{b_t}} - |\alpha| G_\perp. \tag{27}
$$

Suppose that $\boldsymbol{x}_t$ generated by Algorithm 1 satisfies $\nabla f_{S_t}(\boldsymbol{x}_t) = \boldsymbol{0}$ or $\nabla f_S(\boldsymbol{x}_t) = \boldsymbol{0}$. Then, a discussion similar to the one proving Theorem 2.1 under $\nabla f_{S_t}(\boldsymbol{x}_t) = \boldsymbol{0} \vee \nabla f_S(\boldsymbol{x}_t) = \boldsymbol{0}$ ensures that (26) and (27) hold. When $b_t = n$, we have $\boldsymbol{\omega}_t = \nabla \hat{f}_{S,\rho}^{\mathrm{SAM}}(\boldsymbol{x}_t) - \nabla \hat{f}_{S,\rho}^{\mathrm{SAM}}(\boldsymbol{x}_t) + \alpha \nabla f_{S\perp}(\boldsymbol{x}_t) = \alpha \nabla f_{S\perp}(\boldsymbol{x}_t)$, which implies that $\mathbb{E}[\|\boldsymbol{\omega}_t\|_2] = |\alpha| \mathbb{E}[\|\nabla f_{S\perp}(\boldsymbol{x}_t)\|_2]$. $\qquad\square$

## B  GENERAL CONVERGENCE ANALYSIS OF GSAM AND ITS PROOF

**Theorem B.1 ($\epsilon$–approximation of GSAM with an increasing batch size and decaying learning rate)**
*Consider the sequence $(\boldsymbol{x}_t)$ generated by the mini-batch GSAM algorithm (Algorithm 1) with an increasing batch size $b_t \in (0, n]$ and a decaying learning rate $\eta_t \in [\underline{\eta}, \overline{\eta}] \subset [0, +\infty)$ satisfying that there exist positive numbers $H_1(\underline{\eta}, \overline{\eta})$, $H_2(\underline{\eta}, \overline{\eta})$, and $H_3(\underline{\eta}, \overline{\eta})$ such that, for all $T \geq 1$,*

$$\frac{T}{\sum_{t=0}^{T-1} \eta_t} \leq H_1(\underline{\eta}, \overline{\eta}) \quad and \quad \frac{\sum_{t=0}^{T-1} \eta_t^2}{\sum_{t=0}^{T-1} \eta_t} \leq H_2(\underline{\eta}, \overline{\eta}) + \frac{H_3(\underline{\eta}, \overline{\eta})}{T}. \tag{28}$$

*Let us assume that there exists a positive number $G$ such that $\max\{\sup_{t \in \mathbb{N} \cup \{0\}} \|\nabla f_S(\boldsymbol{x}_t + \hat{\boldsymbol{\epsilon}}_{S_t, \rho}(\boldsymbol{x}_t))\|_2, \sup_{t \in \mathbb{N} \cup \{0\}} \|\nabla \hat{f}_{S_t, \rho}^{\mathrm{SAM}}(\boldsymbol{x}_t)\|_2, \sup_{t \in \mathbb{N} \cup \{0\}} \|\nabla \hat{f}_{S, \rho}^{\mathrm{SAM}}(\boldsymbol{x}_t)\|_2, G_\perp\} \leq G$, where $G_\perp := \sup_{t \in \mathbb{N} \cup \{0\}} \|\nabla f_{S_t \perp}(\boldsymbol{x}_t)\|_2 < +\infty$ (Theorem 2.1). Let $\epsilon > 0$ be the precision and let $b_0 > 0$, $\alpha \in \mathbb{R}$, and $\rho \geq 0$ such that*

$$H_1 \leq \frac{\epsilon^2}{12 \sigma C} \left( \frac{\rho G}{\sqrt{b_0}} + \frac{3\sigma}{nb_0} \sum_{i \in [n]} L_i \right)^{-1}, \ (|\alpha| + 1)^2 H_2 \leq \frac{n^3 \epsilon^2}{6 G^2 \sum_{i \in [n]} L_i \{n^2 + 4C(\sum_{i \in [n]} L_i)^2\}},$$

$$\rho(|\alpha| + 1) \leq \frac{n\sqrt{b_0}\epsilon^2}{6G(\sum_{i \in [n]} L_i)(CG\sqrt{b_0} + B\sigma)}, \ \rho^2 \leq \frac{n^2 b_0^2 \epsilon^4}{168 G^2 (n^2 + b_0^2)(\sum_{i \in [n]} L_i)^2}, \tag{29}$$

*where $B$ and $C$ are nonnegative constants. Then, there exists $t_0 \in \mathbb{N}$ such that, for all $T \geq t_0$,*

$$\min_{t \in [0:T-1]} \mathbb{E} \left[ \left\| \nabla \hat{f}_{S, \rho}^{\mathrm{SAM}}(\boldsymbol{x}_t) \right\|_2 \right] \leq \epsilon.$$

Let us start with a brief outline of the proof strategy of Theorem B.1, with an emphasis on the main difficulty that has to be overcome. The flow of our proof is almost the same in Theorem 5.1 of (Zhuang et al., 2022), indicating that GSAM using a decaying learning rate, $\eta_t = \eta_0/\sqrt{t}$, and a perturbation amplitude, $\rho_t = \rho_0/\sqrt{t}$, proportional to $\eta_t$ satisfies

$$\frac{1}{T} \sum_{t=1}^{T} \mathbb{E} \left[ \left\| \nabla \hat{f}_{S, \rho_t}^{\mathrm{SAM}}(\boldsymbol{x}_t) \right\|_2^2 \right] \leq \frac{C_1 + C_2 \log T}{\sqrt{T}},$$

where $C_1$ and $C_2$ are positive constants. First, from the smoothness condition (A1) of $f_S$ and the descent lemma, we prove the inequality (Proposition B.1) that is satisfied for GSAM. Next, using the Cauchy–Schwarz inequality and the triangle inequality, we provide upper bounds of the terms $X_t$ (Proposition B.2), $Y_t$ (Proposition B.3), and $Z_t$ (Proposition B.4) in Proposition B.1. The main issue in Theorem B.1 is to evaluate the full gradient $\nabla \hat{f}_{S, \rho}^{\mathrm{SAM}}(\boldsymbol{x}_t)$ using the mini-batch gradient $\nabla \hat{f}_{S_t, \rho}^{\mathrm{SAM}}(\boldsymbol{x}_t)$. The difficulty comes from the fact that the unbiasedness of $\nabla \hat{f}_{S_t, \rho}^{\mathrm{SAM}}(\boldsymbol{x}_t)$ does not hold (i.e., $\mathbb{E}[\nabla \hat{f}_{S_t, \rho}^{\mathrm{SAM}}(\boldsymbol{x}_t)] \neq \nabla \hat{f}_{S, \rho}^{\mathrm{SAM}}(\boldsymbol{x}_t)$, although (A2)(i) holds). However, we can resolve this issue using Theorem 2.1. In fact, in order to evaluate the upper bound of $X_t$, we can use Theorem 2.1 indicating the upper bound of $\|\hat{\boldsymbol{\omega}}_t\|_2 = \|\nabla \hat{f}_{S, \rho}^{\mathrm{SAM}}(\boldsymbol{x}_t) - \nabla \hat{f}_{S_t, \rho}^{\mathrm{SAM}}(\boldsymbol{x}_t)\|_2$. Another issue that has to be overcome in order to prove Theorem B.1 is to evaluate the upper bound of $\min_{t \in [0:T-1]} \mathbb{E}[\|\nabla \hat{f}_{S, \rho}^{\mathrm{SAM}}(\boldsymbol{x}_t)\|_2^2]$ using a learning rate $\eta_t \in [\underline{\eta}, \overline{\eta}]$. We can resolve this issue by using $\min_{t \in [0:T-1]} \mathbb{E}[\|\nabla \hat{f}_{S, \rho}^{\mathrm{SAM}}(\boldsymbol{x}_t)\|_2^2] \leq \sum_{t=0}^{T-1} \eta_t \mathbb{E}[\|\nabla \hat{f}_{S, \rho}^{\mathrm{SAM}}(\boldsymbol{x}_t)\|_2^2] / \sum_{t=0}^{T-1} \eta_t$. As a result, we can provide an upper bound of $\min_{t \in [0:T-1]} \mathbb{E}[\|\nabla \hat{f}_{S, \rho}^{\mathrm{SAM}}(\boldsymbol{x}_t)\|_2^2]$. Finally, we set $H_1$, $H_2$, $\alpha$, and $\rho$ such that the upper bound of $\min_{t \in [0:T-1]} \mathbb{E}[\|\nabla \hat{f}_{S, \rho}^{\mathrm{SAM}}(\boldsymbol{x}_t)\|_2^2]$ is less than or equal to $\epsilon^2$.

### B.1  LEMMA AND PROPOSITIONS

The following lemma, called the descent lemma, holds.

**Lemma B.1 (Descent lemma)** *(Beck, 2017, Lemma 5.7) Let $f \colon \mathbb{R}^d \to \mathbb{R}$ be $L$–smooth. Then, we have that, for all $\boldsymbol{x}, \boldsymbol{y} \in \mathbb{R}^d$,*

$$f(\boldsymbol{y}) \leq f(\boldsymbol{x}) + \langle \nabla f(\boldsymbol{x}), \boldsymbol{y} - \boldsymbol{x} \rangle_2 + \frac{L}{2} \|\boldsymbol{y} - \boldsymbol{x}\|_2^2.$$

Lemma B.1 leads to the following proposition.

**Proposition B.1** *Under Assumption 2.1, we have that, for all $t \in \mathbb{N} \cup \{0\}$,*

$$f_S(\boldsymbol{x}_{t+1} + \hat{\boldsymbol{\epsilon}}_{S_{t+1},\rho}(\boldsymbol{x}_{t+1}))$$
$$\leq f_S(\boldsymbol{x}_t + \hat{\boldsymbol{\epsilon}}_{S_t,\rho}(\boldsymbol{x}_t))$$
$$+ \eta_t \underbrace{\langle \nabla f_S(\boldsymbol{x}_t + \hat{\boldsymbol{\epsilon}}_{S_t,\rho}(\boldsymbol{x}_t)), \boldsymbol{d}_t \rangle_2}_{X_t} + \underbrace{\langle \nabla f_S(\boldsymbol{x}_t + \hat{\boldsymbol{\epsilon}}_{S_t,\rho}(\boldsymbol{x}_t)), \hat{\boldsymbol{\epsilon}}_{S_{t+1},\rho}(\boldsymbol{x}_{t+1}) - \hat{\boldsymbol{\epsilon}}_{S_t,\rho}(\boldsymbol{x}_t) \rangle_2}_{Y_t}$$
$$+ \frac{\sum_{i \in [n]} L_i}{n} \underbrace{\left\{ \eta_t^2 \|\boldsymbol{d}_t\|_2^2 + \left\| \hat{\boldsymbol{\epsilon}}_{S_{t+1},\rho}(\boldsymbol{x}_{t+1}) - \hat{\boldsymbol{\epsilon}}_{S_t,\rho}(\boldsymbol{x}_t) \right\|_2^2 \right\}}_{Z_t}.$$

*Proof of Proposition B.1:* The $L_i$–smoothness (A1) of $f_i$ and the definition of $f_S$ ensure that, for all $\boldsymbol{x}, \boldsymbol{y} \in \mathbb{R}^d$,

$$\|\nabla f_S(\boldsymbol{x}) - \nabla f_S(\boldsymbol{y})\|_2 = \left\| \frac{1}{n} \sum_{i \in [n]} (\nabla f_i(\boldsymbol{x}) - \nabla f_i(\boldsymbol{y})) \right\|_2 \leq \frac{1}{n} \sum_{i \in [n]} \|\nabla f_i(\boldsymbol{x}) - \nabla f_i(\boldsymbol{y})\|_2$$
$$\leq \frac{1}{n} \sum_{i \in [n]} L_i \|\boldsymbol{x} - \boldsymbol{y}\|_2,$$

which implies that $f_S$ is $(1/n) \sum_{i \in [n]} L_i$–smooth. Lemma B.1 thus guarantees that, for all $t \in \mathbb{N} \cup \{0\}$,

$$f_S(\boldsymbol{x}_{t+1} + \hat{\boldsymbol{\epsilon}}_{S_{t+1},\rho}(\boldsymbol{x}_{t+1}))$$
$$\leq f_S(\boldsymbol{x}_t + \hat{\boldsymbol{\epsilon}}_{S_t,\rho}(\boldsymbol{x}_t)) + \langle \nabla f_S(\boldsymbol{x}_t + \hat{\boldsymbol{\epsilon}}_{S_t,\rho}(\boldsymbol{x}_t)), (\boldsymbol{x}_{t+1} - \boldsymbol{x}_t) + (\hat{\boldsymbol{\epsilon}}_{S_{t+1},\rho}(\boldsymbol{x}_{t+1}) - \hat{\boldsymbol{\epsilon}}_{S_t,\rho}(\boldsymbol{x}_t)) \rangle_2$$
$$+ \frac{\sum_{i \in [n]} L_i}{2n} \left\| (\boldsymbol{x}_{t+1} - \boldsymbol{x}_t) + (\hat{\boldsymbol{\epsilon}}_{S_{t+1},\rho}(\boldsymbol{x}_{t+1}) - \hat{\boldsymbol{\epsilon}}_{S_t,\rho}(\boldsymbol{x}_t)) \right\|_2^2,$$

which, together with $\|\boldsymbol{x} + \boldsymbol{y}\|_2^2 \leq 2(\|\boldsymbol{x}\|_2^2 + \|\boldsymbol{y}\|_2^2)$ and $\boldsymbol{x}_{t+1} - \boldsymbol{x}_t = \eta_t \boldsymbol{d}_t$, implies that

$$f_S(\boldsymbol{x}_{t+1} + \hat{\boldsymbol{\epsilon}}_{S_{t+1},\rho}(\boldsymbol{x}_{t+1}))$$
$$\leq f_S(\boldsymbol{x}_t + \hat{\boldsymbol{\epsilon}}_{S_t,\rho}(\boldsymbol{x}_t))$$
$$+ \eta_t \langle \nabla f_S(\boldsymbol{x}_t + \hat{\boldsymbol{\epsilon}}_{S_t,\rho}(\boldsymbol{x}_t)), \boldsymbol{d}_t \rangle_2 + \langle \nabla f_S(\boldsymbol{x}_t + \hat{\boldsymbol{\epsilon}}_{S_t,\rho}(\boldsymbol{x}_t)), \hat{\boldsymbol{\epsilon}}_{S_{t+1},\rho}(\boldsymbol{x}_{t+1}) - \hat{\boldsymbol{\epsilon}}_{S_t,\rho}(\boldsymbol{x}_t) \rangle_2$$
$$+ \frac{\sum_{i \in [n]} L_i}{n} \left\{ \eta_t^2 \|\boldsymbol{d}_t\|_2^2 + \left\| \hat{\boldsymbol{\epsilon}}_{S_{t+1},\rho}(\boldsymbol{x}_{t+1}) - \hat{\boldsymbol{\epsilon}}_{S_t,\rho}(\boldsymbol{x}_t) \right\|_2^2 \right\},$$

which completes the proof. □

Using Theorem 2.1, we provide an upper bound of $\mathbb{E}[X_t]$.

**Proposition B.2** *Suppose that Assumption 2.1 holds and there exist $G > 0$ and $G_\perp > 0$ such that $\max\{\sup_{t \in \mathbb{N} \cup \{0\}} \|\nabla \hat{f}_{S_t,\rho}^{\mathrm{SAM}}(\boldsymbol{x}_t)\|_2, \sup_{t \in \mathbb{N} \cup \{0\}} \|\nabla \hat{f}_{S,\rho}^{\mathrm{SAM}}(\boldsymbol{x}_t)\|_2\} \leq G$ and $\sup_{t \in \mathbb{N} \cup \{0\}} \|\nabla f_{S_t \perp}(\boldsymbol{x}_t)\|_2 \leq G_\perp$. Then, for all $t \in \mathbb{N} \cup \{0\}$,*

$$\mathbb{E}[X_t] \leq -\mathbb{E}\left[ \left\| \nabla \hat{f}_{S,\rho}^{\mathrm{SAM}}(\boldsymbol{x}_t) \right\|_2^2 \right] + G \sqrt{4\rho^2 \left( \frac{1}{b_t^2} + \frac{1}{n^2} \right) \left( \sum_{i \in [n]} L_i \right)^2 + 2\sigma_t^2}$$
$$+ (G + |\alpha| G_\perp) \frac{\rho B \sigma}{n \sqrt{b_t}} \sum_{i \in [n]} L_i,$$

*where $\sigma_t^2 := \mathbb{E}[\|\nabla f_{S_t}(\boldsymbol{x}_t) - \nabla f_S(\boldsymbol{x}_t)\|_2^2] \leq \sigma^2 / b_t$ and $B \geq 0$ is a constant.*

*Proof:* Let $t \in \mathbb{N} \cup \{0\}$ and $b_t < n$. The definition of $\boldsymbol{d}_t = -(\nabla \hat{f}_{S_t,\rho}^{\mathrm{SAM}}(\boldsymbol{x}_t) - \alpha \nabla f_{S_t \perp}(\boldsymbol{x}_t))$ implies that

$$X_t = \underbrace{-\left\langle \nabla f_S(\boldsymbol{x}_t + \hat{\boldsymbol{\epsilon}}_{S_t,\rho}(\boldsymbol{x}_t)), \nabla \hat{f}_{S_t,\rho}^{\mathrm{SAM}}(\boldsymbol{x}_t) \right\rangle_2}_{X_{t,1}} + \underbrace{\alpha \left\langle \nabla f_S(\boldsymbol{x}_t + \hat{\boldsymbol{\epsilon}}_{S_t,\rho}(\boldsymbol{x}_t)), \nabla f_{S_t \perp}(\boldsymbol{x}_t) \right\rangle_2}_{X_{t,2}}. \quad (30)$$

Then, we have

$$
X_{t,1} = -\Bigg\{ \left\| \nabla \hat{f}_{S,\rho}^{\mathrm{SAM}}(\boldsymbol{x}_t) \right\|_2^2 + \left\langle \nabla f_S(\boldsymbol{x}_t + \hat{\boldsymbol{\epsilon}}_{S_t,\rho}(\boldsymbol{x}_t)) - \nabla \hat{f}_{S,\rho}^{\mathrm{SAM}}(\boldsymbol{x}_t), \nabla \hat{f}_{S_t,\rho}^{\mathrm{SAM}}(\boldsymbol{x}_t) \right\rangle_2
$$

$$
+ \left\langle \nabla \hat{f}_{S,\rho}^{\mathrm{SAM}}(\boldsymbol{x}_t), \underbrace{\nabla \hat{f}_{S_t,\rho}^{\mathrm{SAM}}(\boldsymbol{x}_t) - \nabla \hat{f}_{S,\rho}^{\mathrm{SAM}}(\boldsymbol{x}_t)}_{-\hat{\boldsymbol{\omega}}_t} \right\rangle_2 \Bigg\}
$$

$$
\leq - \left\| \nabla \hat{f}_{S,\rho}^{\mathrm{SAM}}(\boldsymbol{x}_t) \right\|_2^2 + \underbrace{\left\| \nabla f_S(\boldsymbol{x}_t + \hat{\boldsymbol{\epsilon}}_{S_t,\rho}(\boldsymbol{x}_t)) - \nabla \hat{f}_{S,\rho}^{\mathrm{SAM}}(\boldsymbol{x}_t) \right\|_2}_{X_{t,3}} \left\| \nabla \hat{f}_{S_t,\rho}^{\mathrm{SAM}}(\boldsymbol{x}_t) \right\|_2 \tag{31}
$$

$$
+ \left\| \nabla \hat{f}_{S,\rho}^{\mathrm{SAM}}(\boldsymbol{x}_t) \right\|_2 \left\| \hat{\boldsymbol{\omega}}_t \right\|_2,
$$

where the second inequality comes from the Cauchy–Schwarz inequality. Suppose that $\nabla f_{S_t}(\boldsymbol{x}_t) \neq \mathbf{0}$ and $\nabla f_S(\boldsymbol{x}_t) \neq \mathbf{0}$. The $(1/n)\sum_{i \in [n]} L_i$–smoothness of $f_S$ implies that

$$
X_{t,3} = \left\| \nabla f_S \left( \boldsymbol{x}_t + \rho \frac{\nabla f_{S_t}(\boldsymbol{x}_t)}{\|\nabla f_{S_t}(\boldsymbol{x}_t)\|_2} \right) - \nabla f_S \left( \boldsymbol{x}_t + \rho \frac{\nabla f_S(\boldsymbol{x}_t)}{\|\nabla f_S(\boldsymbol{x}_t)\|_2} \right) \right\|_2
$$

$$
\leq \frac{\rho}{n} \sum_{i \in [n]} L_i \left\| \frac{\nabla f_{S_t}(\boldsymbol{x}_t)}{\|\nabla f_{S_t}(\boldsymbol{x}_t)\|_2} - \frac{\nabla f_S(\boldsymbol{x}_t)}{\|\nabla f_S(\boldsymbol{x}_t)\|_2} \right\|_2.
$$

The discussion in (Zhuang et al., 2022, Pages 15 and 16) implies there exists $B_t \geq 0$ such that

$$
\left\| \frac{\nabla f_{S_t}(\boldsymbol{x}_t)}{\|\nabla f_{S_t}(\boldsymbol{x}_t)\|_2} - \frac{\nabla f_S(\boldsymbol{x}_t)}{\|\nabla f_S(\boldsymbol{x}_t)\|_2} \right\|_2 \leq B_t \left\| \nabla f_{S_t}(\boldsymbol{x}_t) - \nabla f_S(\boldsymbol{x}_t) \right\|_2 \tag{32}
$$

Let $B := \sup_{t \in \mathbb{N} \cup \{0\}} B_t$. Then, Proposition A.1 ensures that

$$
\mathbb{E}[X_{t,3}] \leq \frac{\rho B \sigma}{n \sqrt{b_t}} \sum_{i \in [n]} L_i. \tag{33}
$$

Suppose that $\nabla f_{S_t}(\boldsymbol{x}_t) = \mathbf{0}$ or $\nabla f_S(\boldsymbol{x}_t) = \mathbf{0}$. Let $\nabla f_{S_t}(\boldsymbol{x}_t) = \mathbf{0}$. The $(1/n)\sum_{i \in [n]} L_i$–smoothness of $f_S$ ensures that

$$
X_{t,3} = \left\| \nabla f_S \left( \boldsymbol{x}_t + \boldsymbol{u} \right) - \nabla f_S \left( \boldsymbol{x}_t + \rho \frac{\nabla f_S(\boldsymbol{x}_t)}{\|\nabla f_S(\boldsymbol{x}_t)\|_2} \right) \right\|_2 \leq \frac{1}{n} \sum_{i \in [n]} L_i \left\| \boldsymbol{u} - \rho \frac{\nabla f_S(\boldsymbol{x}_t)}{\|\nabla f_S(\boldsymbol{x}_t)\|_2} \right\|_2,
$$

which, together with $\|\boldsymbol{u}\|_2 \leq \rho$, implies there exists $C_t \geq 0$ such that

$$
X_{t,3} \leq \frac{\rho C_t}{n} \sum_{i \in [n]} L_i \left\| \frac{\nabla f_{S_t}(\boldsymbol{x}_t)}{\|\nabla f_{S_t}(\boldsymbol{x}_t)\|_2} - \frac{\nabla f_S(\boldsymbol{x}_t)}{\|\nabla f_S(\boldsymbol{x}_t)\|_2} \right\|_2
$$

Hence, Proposition A.1 implies that (33) holds. A discussion similar to the case where $\nabla f_{S_t}(\boldsymbol{x}_t) = \mathbf{0}$ ensures that (33) holds for $\nabla f_S(\boldsymbol{x}_t) = \mathbf{0}$. Taking the total expectation on both sides of (31), together with (33) and Theorem 2.1, yields

$$
\mathbb{E}[X_{t,1}] \leq -\mathbb{E}\left[ \left\| \nabla \hat{f}_{S,\rho}^{\mathrm{SAM}}(\boldsymbol{x}_t) \right\|_2^2 \right] + G \sqrt{4\rho^2 \left( \frac{1}{b_t^2} + \frac{1}{n^2} \right) \left( \sum_{i \in [n]} L_i \right)^2 + 2\sigma_t^2}
$$

$$
+ \frac{\rho B G \sigma}{n \sqrt{b_t}} \sum_{i \in [n]} L_i. \tag{34}
$$

The Cauchy–Schwarz inequality implies that

$$
X_{t,2} = \alpha \left\langle \nabla f_S(\boldsymbol{x}_t + \hat{\boldsymbol{\epsilon}}_{S_t,\rho}(\boldsymbol{x}_t)) - \nabla \hat{f}_{S,\rho}^{\mathrm{SAM}}(\boldsymbol{x}_t) + \nabla \hat{f}_{S,\rho}^{\mathrm{SAM}}(\boldsymbol{x}_t), \nabla f_{S_t \perp}(\boldsymbol{x}_t) \right\rangle_2
$$

$$
\leq |\alpha| X_{t,3} \left\| \nabla f_{S_t \perp}(\boldsymbol{x}_t) \right\|_2 + \alpha \left\langle \nabla \hat{f}_{S,\rho}^{\mathrm{SAM}}(\boldsymbol{x}_t), \nabla f_{S_t \perp}(\boldsymbol{x}_t) \right\rangle_2
$$

$$\leq |\alpha| G_\perp X_{t,3} + \alpha \left\langle \nabla \hat{f}_{S,\rho}^{\text{SAM}}(\boldsymbol{x}_t), \nabla f_{S_t\perp}(\boldsymbol{x}_t) \right\rangle_2,$$

which, together with $\mathbb{E}_{\boldsymbol{\xi}_t}[\nabla f_{S_t\perp}(\boldsymbol{x}_t)|\boldsymbol{\xi}_{t-1}] = \nabla f_{S\perp}(\boldsymbol{x}_t)$, $\langle \nabla \hat{f}_{S,\rho}^{\text{SAM}}(\boldsymbol{x}_t), \nabla f_{S\perp}(\boldsymbol{x}_t)\rangle_2 = 0$, and (33), implies that

$$\mathbb{E}[X_{t,2}] \leq \frac{|\alpha|\rho B G_\perp \sigma}{n\sqrt{b_t}} \sum_{i\in[n]} L_i. \tag{35}$$

Accordingly, (30), (34), and (35) guarantee that

$$\mathbb{E}[X_t] \leq -\mathbb{E}\left[\left\|\nabla \hat{f}_{S,\rho}^{\text{SAM}}(\boldsymbol{x}_t)\right\|_2^2\right] + G\sqrt{4\rho^2\left(\frac{1}{b_t^2} + \frac{1}{n^2}\right)\left(\sum_{i\in[n]} L_i\right)^2 + 2\sigma_t^2}$$

$$+ (G + |\alpha|G_\perp)\frac{\rho B\sigma}{n\sqrt{b_t}} \sum_{i\in[n]} L_i,$$

which completes the proof. $\qquad\square$

**Proposition B.3** *Suppose that the assumptions in Proposition B.2 hold and there exists $G > 0$ such that $\max\{\sup_{t\in\mathbb{N}\cup\{0\}} \|\nabla f_S(\boldsymbol{x}_t + \hat{\boldsymbol{\epsilon}}_{S_t,\rho}(\boldsymbol{x}_t))\|_2, \sup_{t\in\mathbb{N}\cup\{0\}} \|\nabla \hat{f}_{S_t,\rho}^{\text{SAM}}(\boldsymbol{x}_t)\|_2, \sup_{t\in\mathbb{N}\cup\{0\}} \|\nabla \hat{f}_{S,\rho}^{\text{SAM}}(\boldsymbol{x}_t)\|_2, G_\perp\} \leq G$. Then, for all $t \in \mathbb{N}\cup\{0\}$,*

$$\mathbb{E}[Y_t] \leq \rho C G\left\{\frac{\eta_t(|\alpha|+1)G}{n}\sum_{i\in[n]} L_i + \frac{2\sigma}{\sqrt{b_t}}\right\},$$

*where $C \geq 0$ is a constant.*

*Proof:* Let $t \in \mathbb{N}\cup\{0\}$. The Cauchy–Schwarz inequality ensures that

$$Y_t \leq G\left\|\hat{\boldsymbol{\epsilon}}_{S_{t+1},\rho}(\boldsymbol{x}_{t+1}) - \hat{\boldsymbol{\epsilon}}_{S_t,\rho}(\boldsymbol{x}_t)\right\|_2 =: GY_{t,1}. \tag{36}$$

Suppose that $\nabla f_{S_{t+1}}(\boldsymbol{x}_{t+1}) \neq \boldsymbol{0}$ and $\nabla f_{S_t}(\boldsymbol{x}_t) \neq \boldsymbol{0}$. The discussion in (Zhuang et al., 2022, Pages 15 and 16) (see (32)) implies that there exists $C_t \geq 0$ such that

$$Y_{t,1} = \rho\left\|\frac{\nabla f_{S_{t+1}}(\boldsymbol{x}_{t+1})}{\|\nabla f_{S_{t+1}}(\boldsymbol{x}_{t+1})\|_2} - \frac{\nabla f_{S_t}(\boldsymbol{x}_t)}{\|\nabla f_{S_t}(\boldsymbol{x}_t)\|_2}\right\|_2 \leq \rho C_t\left\|\nabla f_{S_{t+1}}(\boldsymbol{x}_{t+1}) - \nabla f_{S_t}(\boldsymbol{x}_t)\right\|_2. \tag{37}$$

Let $C := \sup_{t\in\mathbb{N}\cup\{0\}} C_t$. The triangle inequality gives

$$\left\|\nabla f_{S_{t+1}}(\boldsymbol{x}_{t+1}) - \nabla f_{S_t}(\boldsymbol{x}_t)\right\|_2$$
$$\leq \left\|\nabla f_{S_{t+1}}(\boldsymbol{x}_{t+1}) - \nabla f_S(\boldsymbol{x}_{t+1})\right\|_2 + \left\|\nabla f_S(\boldsymbol{x}_{t+1}) - \nabla f_S(\boldsymbol{x}_t)\right\|_2 + \left\|\nabla f_S(\boldsymbol{x}_t) - \nabla f_{S_t}(\boldsymbol{x}_t)\right\|_2,$$

which, together with the $(1/n)\sum_{i\in[n]} L_i$–smoothness of $f_S$, $\boldsymbol{x}_{t+1} - \boldsymbol{x}_t = \eta_t \boldsymbol{d}_t$, (36), and (37), implies that

$$Y_{t,1} \leq \rho C\left\{\frac{\eta_t}{n}\sum_{i\in[n]} L_i\|\boldsymbol{d}_t\|_2 + \left\|\nabla f_{S_{t+1}}(\boldsymbol{x}_{t+1}) - \nabla f_S(\boldsymbol{x}_{t+1})\right\|_2 + \left\|\nabla f_{S_t}(\boldsymbol{x}_t) - \nabla f_S(\boldsymbol{x}_t)\right\|_2\right\}.$$

Moreover, the Cauchy–Schwarz inequality and the definitions of $G$ and $G_\perp$ ensure that

$$\|\boldsymbol{d}_t\|_2^2 = \left\|\nabla \hat{f}_{S_t,\rho}^{\text{SAM}}(\boldsymbol{x}_t) - \alpha\nabla f_{S_t\perp}(\boldsymbol{x}_t)\right\|_2^2$$
$$= \left\|\nabla \hat{f}_{S_t,\rho}^{\text{SAM}}(\boldsymbol{x}_t)\right\|_2^2 - 2\alpha\left\langle\nabla \hat{f}_{S_t,\rho}^{\text{SAM}}(\boldsymbol{x}_t), \nabla f_{S_t\perp}(\boldsymbol{x}_t)\right\rangle_2 + |\alpha|^2\left\|\nabla f_{S_t\perp}(\boldsymbol{x}_t)\right\|_2^2 \tag{38}$$
$$\leq G^2 + 2|\alpha|GG_\perp + |\alpha|^2 G_\perp^2 \leq (|\alpha|+1)^2 G^2.$$

Accordingly, we have

$$Y_{t,1} \leq \rho C\left\{\frac{\eta_t(|\alpha|+1)G}{n}\sum_{i\in[n]} L_i + \left\|\nabla f_{S_{t+1}}(\boldsymbol{x}_{t+1}) - \nabla f_S(\boldsymbol{x}_{t+1})\right\|_2 + \left\|\nabla f_{S_t}(\boldsymbol{x}_t) - \nabla f_S(\boldsymbol{x}_t)\right\|_2\right\},$$

which, together with Proposition A.1, guarantees that

$$\mathbb{E}[Y_{t,1}] \le \rho C \left\{ \frac{\eta_t(|\alpha|+1)G}{n} \sum_{i\in[n]} L_i + \frac{2\sigma}{\sqrt{b_t}} \right\}. \tag{39}$$

Hence, from (36),

$$\mathbb{E}[Y_t] \le \rho C G \left\{ \frac{\eta_t(|\alpha|+1)G}{n} \sum_{i\in[n]} L_i + \frac{2\sigma}{\sqrt{b_t}} \right\}.$$

We can show that Proposition B.3 holds for the case where $\nabla f_{S_{t+1}}(\boldsymbol{x}_{t+1}) = \boldsymbol{0}$ or $\nabla f_{S_t}(\boldsymbol{x}_t) = \boldsymbol{0}$ by proving Proposition B.2. □

**Proposition B.4** *Suppose that the assumptions in Proposition B.3 hold. Then, for all $t \in \mathbb{N} \cup \{0\}$,*

$$\mathbb{E}[Z_t] \le \eta_t^2(|\alpha|+1)^2 G^2 \left\{ 1 + \frac{4C}{n^2} \left( \sum_{i\in[n]} L_i \right)^2 \right\} + \frac{6C\sigma^2}{b_t}.$$

*Proof:* Let $t \in \mathbb{N} \cup \{0\}$. From (38), we have

$$\eta_t^2 \mathbb{E}[\|\boldsymbol{d}_t\|_2] \le \eta_t^2(|\alpha|+1)^2 G^2.$$

Suppose that $\nabla f_{S_{t+1}}(\boldsymbol{x}_{t+1}) \ne \boldsymbol{0}$ and $\nabla f_{S_t}(\boldsymbol{x}_t) \ne \boldsymbol{0}$. Then, from $\|\boldsymbol{x}+\boldsymbol{y}\|_2^2 \le 2(\|\boldsymbol{x}\|_2^2 + \|\boldsymbol{y}\|_2^2)$,

$$\left\| \nabla f_{S_{t+1}}(\boldsymbol{x}_{t+1}) - \nabla f_{S_t}(\boldsymbol{x}_t) \right\|_2^2$$
$$\le 2 \left\| \nabla f_{S_{t+1}}(\boldsymbol{x}_{t+1}) - \nabla f_S(\boldsymbol{x}_{t+1}) \right\|_2^2 + 4 \left\| \nabla f_S(\boldsymbol{x}_{t+1}) - \nabla f_S(\boldsymbol{x}_t) \right\|_2^2 + 4 \left\| \nabla f_S(\boldsymbol{x}_t) - \nabla f_{S_t}(\boldsymbol{x}_t) \right\|_2^2.$$

A discussion similar to the one showing (39) ensures that

$$\mathbb{E}[Y_{t,1}^2] = \mathbb{E}\left[ \left\| \hat{\boldsymbol{\epsilon}}_{S_{t+1},\rho}(\boldsymbol{x}_{t+1}) - \hat{\boldsymbol{\epsilon}}_{S_t,\rho}(\boldsymbol{x}_t) \right\|_2^2 \right] \le 2C \left\{ \frac{2\eta_t^2(|\alpha|+1)^2 G^2}{n^2} \left( \sum_{i\in[n]} L_i \right)^2 + \frac{3\sigma^2}{b_t} \right\}.$$

The above inequality holds for the case where $\nabla f_{S_{t+1}}(\boldsymbol{x}_{t+1}) = \boldsymbol{0}$ or $\nabla f_{S_t}(\boldsymbol{x}_t) = \boldsymbol{0}$ by an argument similar to the one used to prove Proposition B.2. Hence,

$$\mathbb{E}[Z_t] \le \eta_t^2(|\alpha|+1)^2 G^2 + 2C \left\{ \frac{2\eta_t^2(|\alpha|+1)^2 G^2}{n^2} \left( \sum_{i\in[n]} L_i \right)^2 + \frac{3\sigma^2}{b_t} \right\}$$

$$= \eta_t^2(|\alpha|+1)^2 G^2 \left\{ 1 + \frac{4C}{n^2} \left( \sum_{i\in[n]} L_i \right)^2 \right\} + \frac{6C\sigma^2}{b_t},$$

which completes the proof. □

*Proof of Theorem B.1:* Let us define $F_\rho(t) := f_S(\boldsymbol{x}_t + \hat{\boldsymbol{\epsilon}}_{S_t,\rho}(\boldsymbol{x}_t))$. From Proposition B.1, Proposition B.2, Proposition B.3, and Proposition B.4, for all $t \in \mathbb{N} \cup \{0\}$, we have

$$\mathbb{E}[F_\rho(t+1)] \le \mathbb{E}[F_\rho(t)] + \eta_t \mathbb{E}[X_t] + \mathbb{E}[Y_t] + \frac{\sum_{i\in[n]} L_i}{n} \mathbb{E}[Z_t]$$

$$\le \mathbb{E}[F_\rho(t)] - \eta_t \mathbb{E}\left[ \left\| \nabla \hat{f}_{S,\rho}^{\text{SAM}}(\boldsymbol{x}_t) \right\|_2^2 \right] + \eta_t G \sqrt{ 4\rho^2 \left( \frac{1}{b_t^2} + \frac{1}{n^2} \right) \left( \sum_{i\in[n]} L_i \right)^2 + 2\sigma_t^2 }$$

$$+ \eta_t(|\alpha|+1) \frac{\rho B G \sigma}{n\sqrt{b_t}} \sum_{i\in[n]} L_i + \rho C G \left\{ \frac{\eta_t(|\alpha|+1)G}{n} \sum_{i\in[n]} L_i + \frac{2\sigma}{\sqrt{b_t}} \right\}$$

$$+ \frac{\sum_{i \in [n]} L_i}{n} \left[ \eta_t^2 (|\alpha| + 1)^2 G^2 \left\{ 1 + \frac{4C}{n^2} \left( \sum_{i \in [n]} L_i \right)^2 \right\} + \frac{6C\sigma^2}{b_t} \right],$$

which implies that

$$\eta_t \mathbb{E} \left[ \left\| \nabla \hat{f}_{S,\rho}^{\mathrm{SAM}}(\boldsymbol{x}_t) \right\|_2^2 \right] \leq (\mathbb{E}[F_\rho(t)] - \mathbb{E}[F_\rho(t+1)]) + 2\sigma C \left( \frac{\rho G}{\sqrt{b_t}} + \frac{3\sigma}{nb_t} \sum_{i \in [n]} L_i \right)$$

$$+ \frac{\eta_t^2 (|\alpha| + 1)^2 G^2}{n} \sum_{i \in [n]} L_i \left\{ 1 + \frac{4C}{n^2} \left( \sum_{i \in [n]} L_i \right)^2 \right\} \tag{40}$$

$$+ \eta_t G \sqrt{4\rho^2 \left( \frac{1}{b_t^2} + \frac{1}{n^2} \right) \left( \sum_{i \in [n]} L_i \right)^2 + 2\sigma_t^2}$$

$$+ \eta_t \frac{\rho(|\alpha| + 1)G}{n} \sum_{i \in [n]} L_i \left( CG + \frac{B\sigma}{\sqrt{b_t}} \right).$$

Let $\epsilon > 0$. From $g(b_t) = \sigma_t^2 := \mathbb{E}[\|\nabla f_{S_t}(\boldsymbol{x}_t) - \nabla f_S(\boldsymbol{x}_t)\|_2^2] \leq \sigma^2/b_t \ (t \in \mathbb{N} \cup \{0\})$ (see Proposition A.1 and (Freund, 1971, Theorem 8.6)) and $g(n) = 0$, the sequence $(b_t)$ of increasing batch sizes implies that there exists $t_0 \in \mathbb{N}$ such that, for all $t \geq t_0$,

$$2\sigma_t^2 \leq \frac{\epsilon^4}{49G^2}.$$

Let $T \geq t_0 + 1$. Summing the above inequality from $t = 0$ to $t = T - 1$, together with $b_0 \leq b_t$ and $\eta_t \leq \overline{\eta} \ (t \in \mathbb{N} \cup \{0\})$, ensures that

$$\sum_{t=0}^{T-1} \eta_t \mathbb{E} \left[ \left\| \nabla \hat{f}_{S,\rho}^{\mathrm{SAM}}(\boldsymbol{x}_t) \right\|_2^2 \right] \leq (\mathbb{E}[F_\rho(0)] - f_S^\star) + 2\sigma C \left( \frac{\rho G}{\sqrt{b_0}} + \frac{3\sigma}{nb_0} \sum_{i \in [n]} L_i \right) T$$

$$+ \frac{(|\alpha| + 1)^2 G^2}{n} \sum_{i \in [n]} L_i \left\{ 1 + \frac{4C}{n^2} \left( \sum_{i \in [n]} L_i \right)^2 \right\} \sum_{t=0}^{T-1} \eta_t^2$$

$$+ G \sqrt{4\rho^2 \left( \frac{1}{b_0^2} + \frac{1}{n^2} \right) \left( \sum_{i \in [n]} L_i \right)^2 + \frac{2\sigma^2}{b_0}} t_0 \overline{\eta}$$

$$+ G \sqrt{4\rho^2 \left( \frac{1}{b_0^2} + \frac{1}{n^2} \right) \left( \sum_{i \in [n]} L_i \right)^2 + \frac{\epsilon^4}{49G^2}} \sum_{t=t_0}^{T-1} \eta_t$$

$$+ \frac{\rho(|\alpha| + 1)G}{n} \sum_{i \in [n]} L_i \left( CG + \frac{B\sigma}{\sqrt{b_0}} \right) \sum_{t=0}^{T-1} \eta_t,$$

where $f_S^\star$ is the minimum value of $f_S$ over $\mathbb{R}^d$. Since we have that

$$\min_{t \in [0:T-1]} \mathbb{E} \left[ \left\| \nabla \hat{f}_{S,\rho}^{\mathrm{SAM}}(\boldsymbol{x}_t) \right\|_2^2 \right] \leq \frac{\sum_{t=0}^{T-1} \eta_t \mathbb{E} \left[ \left\| \nabla \hat{f}_{S,\rho}^{\mathrm{SAM}}(\boldsymbol{x}_t) \right\|_2^2 \right]}{\sum_{t=0}^{T-1} \eta_t},$$

we also have that

$$\min_{t \in [0:T-1]} \mathbb{E} \left[ \left\| \nabla \hat{f}_{S,\rho}^{\mathrm{SAM}}(\boldsymbol{x}_t) \right\|_2^2 \right] \leq \frac{\mathbb{E}[F_\rho(0)] - f_S^\star}{\sum_{t=0}^{T-1} \eta_t} + 2\sigma C \left( \frac{\rho G}{\sqrt{b_0}} + \frac{3\sigma}{nb_0} \sum_{i \in [n]} L_i \right) \frac{T}{\sum_{t=0}^{T-1} \eta_t}$$

$$+ \frac{(|\alpha| + 1)^2 G^2}{n} \sum_{i \in [n]} L_i \left\{ 1 + \frac{4C}{n^2} \left( \sum_{i \in [n]} L_i \right)^2 \right\} \frac{\sum_{t=0}^{T-1} \eta_t^2}{\sum_{t=0}^{T-1} \eta_t}$$

$$+ G\sqrt{4\rho^2\left(\frac{1}{b_0^2} + \frac{1}{n^2}\right)\left(\sum_{i\in[n]} L_i\right)^2 + \frac{2\sigma^2}{b_0}}\frac{t_0\overline{\eta}}{\sum_{t=0}^{T-1}\eta_t}$$

$$+ G\sqrt{4\rho^2\left(\frac{1}{b_0^2} + \frac{1}{n^2}\right)\left(\sum_{i\in[n]} L_i\right)^2 + \frac{\epsilon^4}{49G^2}}$$

$$+ \frac{\rho(|\alpha| + 1)G}{n}\sum_{i\in[n]} L_i\left(CG + \frac{B\sigma}{\sqrt{b_0}}\right). \tag{41}$$

From (28), i.e.,

$$\frac{T}{\sum_{t=0}^{T-1}\eta_t} \le H_1(\underline{\eta}, \overline{\eta}) \quad \text{and} \quad \frac{\sum_{t=0}^{T-1}\eta_t^2}{\sum_{t=0}^{T-1}\eta_t} \le H_2(\underline{\eta}, \overline{\eta}) + \frac{H_3(\underline{\eta}, \overline{\eta})}{T},$$

we have that

$$\min_{t\in[0:T-1]} \mathbb{E}\left[\left\|\nabla \hat{f}_{S,\rho}^{\mathrm{SAM}}(\boldsymbol{x}_t)\right\|_2^2\right]$$

$$\le \underbrace{\frac{H_1(\mathbb{E}[F_\rho(0)] - f_S^\star)}{T} + GH_1\sqrt{4\rho^2\left(\frac{1}{b_0^2} + \frac{1}{n^2}\right)\left(\sum_{i\in[n]} L_i\right)^2 + \frac{2\sigma^2}{b_0}}\frac{t_0\overline{\eta}}{T}}_{U_1 \le \frac{\epsilon^2}{6}}$$

$$+ \underbrace{\frac{(|\alpha| + 1)^2 G^2}{n}\sum_{i\in[n]} L_i\left\{1 + \frac{4C}{n^2}\left(\sum_{i\in[n]} L_i\right)^2\right\}\frac{H_3}{T}}_{U_2 \le \frac{\epsilon^2}{6}}$$

$$+ \underbrace{\left(\frac{\rho G}{\sqrt{b_0}} + \frac{3\sigma}{nb_0}\sum_{i\in[n]} L_i\right)2\sigma CH_1}_{U_3 \le \frac{\epsilon^2}{6}} + \underbrace{\frac{(|\alpha| + 1)^2 G^2 H_2}{n}\sum_{i\in[n]} L_i\left\{1 + \frac{4C}{n^2}\left(\sum_{i\in[n]} L_i\right)^2\right\}}_{U_4 \le \frac{\epsilon^2}{6}}$$

$$+ \underbrace{\frac{\rho(|\alpha| + 1)G}{n}\sum_{i\in[n]} L_i\left(CG + \frac{B\sigma}{\sqrt{b_0}}\right)}_{U_5 \le \frac{\epsilon^2}{6}} + \underbrace{G\sqrt{4\rho^2\left(\frac{1}{b_0^2} + \frac{1}{n^2}\right)\left(\sum_{i\in[n]} L_i\right)^2 + \frac{\epsilon^4}{49G^2}}}_{U_6 \le \frac{\epsilon^2}{6}}.$$

It is guaranteed that there exists $t_1 \in \mathbb{N}$ such that, for all $T \ge \max\{t_0, t_1\}$, $U_1 \le \frac{\epsilon^2}{6}$ and $U_2 \le \frac{\epsilon^2}{6}$. Moreover, if (29) holds, i.e.,

$$H_1 \le \frac{\epsilon^2}{12\sigma C}\left(\frac{\rho G}{\sqrt{b_0}} + \frac{3\sigma}{nb_0}\sum_{i\in[n]} L_i\right)^{-1}, \quad (|\alpha| + 1)^2 H_2 \le \frac{n^3\epsilon^2}{6G^2\sum_{i\in[n]} L_i\{n^2 + 4C(\sum_{i\in[n]} L_i)^2\}},$$

$$\rho(|\alpha| + 1) \le \frac{n\sqrt{b_0}\epsilon^2}{6G(\sum_{i\in[n]} L_i)(CG\sqrt{b_0} + B\sigma)}, \quad \rho^2 \le \frac{n^2 b_0^2\epsilon^4}{168G^2(n^2 + b_0^2)(\sum_{i\in[n]} L_i)^2},$$

then $U_i \le \frac{\epsilon^2}{6}$ ($i = 3, 4, 5, 6$), i.e.,

$$\min_{t\in[0:T-1]} \mathbb{E}\left[\left\|\nabla \hat{f}_{S,\rho}^{\mathrm{SAM}}(\boldsymbol{x}_t)\right\|_2\right] \le \epsilon. \tag{42}$$

This completes the proof. $\qquad\qquad\square$

## B.2 PROOF OF THEOREM 2.3

Let $\eta_t = \eta > 0$. Then, we have

$$\frac{T}{\sum_{t=0}^{T-1} \eta_t} = \frac{1}{\eta} =: H_1 \text{ and } \frac{\sum_{t=0}^{T-1} \eta_t^2}{\sum_{t=0}^{T-1} \eta_t} = \eta =: H_2,$$

which implies that (28) with $H_3 = 0$ holds. Hence, from (29), the assertion in Theorem 2.3 holds.
□

## B.3 PROOF OF THEOREM 2.4

We can prove the following corollary by using Theorem B.1.

**Corollary B.1 ($\epsilon$–approximation of GSAM with a constant batch size and decaying learning rate)** *Consider the sequence $(\boldsymbol{x}_t)$ generated by the mini-batch GSAM algorithm (Algorithm 1) with a constant batch size $b \in (0, n]$ and a decaying learning rate $\eta_t \in [\underline{\eta}, \overline{\eta}] \subset [0, +\infty)$ satisfying that there exist positive numbers $H_1(\underline{\eta}, \overline{\eta})$, $H_2(\underline{\eta}, \overline{\eta})$, and $H_3(\underline{\eta}, \overline{\eta})$ such that, for all $T \geq 1$, (28) holds. We will assume that there exists a positive number $G$ such that $\max\{\sup_{t \in \mathbb{N} \cup \{0\}} \|\nabla f_S(\boldsymbol{x}_t + \hat{\epsilon}_{S_t,\rho}(\boldsymbol{x}_t))\|_2, \sup_{t \in \mathbb{N} \cup \{0\}} \|\nabla \hat{f}_{S_t,\rho}^{\mathrm{SAM}}(\boldsymbol{x}_t)\|_2, \sup_{t \in \mathbb{N} \cup \{0\}} \|\nabla \hat{f}_{S,\rho}^{\mathrm{SAM}}(\boldsymbol{x}_t)\|_2, G_\perp\} \leq G$, where $G_\perp := \sup_{t \in \mathbb{N} \cup \{0\}} \|\nabla f_{S_t \perp}(\boldsymbol{x}_t)\|_2 < +\infty$ (Theorem 2.1). Let $\epsilon > 0$ be the precision and let $b_0 > 0$, $\alpha \in \mathbb{R}$, and $\rho \geq 0$ such that*

$$H_1 \leq \frac{\epsilon^2}{12\sigma C} \left( \frac{\rho G}{\sqrt{b}} + \frac{3\sigma}{nb} \sum_{i \in [n]} L_i \right)^{-1}, \quad (|\alpha| + 1)^2 H_2 \leq \frac{n^3 \epsilon^2}{6G^2 \sum_{i \in [n]} L_i \{n^2 + 4C(\sum_{i \in [n]} L_i)^2\}},$$

$$\rho(|\alpha| + 1) \leq \frac{n\sqrt{b}\epsilon^2}{6G(\sum_{i \in [n]} L_i)(CG\sqrt{b} + B\sigma)}, \quad \rho^2 \leq \frac{n^2 b^2 \epsilon^4}{168 G^2 (n^2 + b^2)(\sum_{i \in [n]} L_i)^2}, \tag{43}$$

*where $B$ and $C$ are nonnegative constants. Then, there exists $t_0 \in \mathbb{N}$ such that, for all $T \geq t_0$,*

$$\min_{t \in [0:T-1]} \mathbb{E}\left[ \left\| \nabla \hat{f}_{S,\rho}^{\mathrm{SAM}}(\boldsymbol{x}_t) \right\|_2 \right] \leq \epsilon.$$

*Proof:* Let $b_t = b$ ($t \in \mathbb{N} \cup \{0\}$). Using inequality (40) that was used to prove Theorem B.1, we have that, for all $t \in \mathbb{N} \cup \{0\}$,

$$\eta_t \mathbb{E}\left[ \left\| \nabla \hat{f}_{S,\rho}^{\mathrm{SAM}}(\boldsymbol{x}_t) \right\|_2^2 \right] \leq (\mathbb{E}[F_\rho(t)] - \mathbb{E}[F_\rho(t+1)]) + 2\sigma C \left( \frac{\rho G}{\sqrt{b}} + \frac{3\sigma}{nb} \sum_{i \in [n]} L_i \right)$$

$$+ \frac{\eta_t^2 (|\alpha| + 1)^2 G^2}{n} \sum_{i \in [n]} L_i \left\{ 1 + \frac{4C}{n^2} \left( \sum_{i \in [n]} L_i \right)^2 \right\}$$

$$+ \eta_t G \sqrt{4\rho^2 \left( \frac{1}{b^2} + \frac{1}{n^2} \right) \left( \sum_{i \in [n]} L_i \right)^2 + 2\sigma_t^2}$$

$$+ \eta_t \frac{\rho(|\alpha| + 1)G}{n} \sum_{i \in [n]} L_i \left( CG + \frac{B\sigma}{\sqrt{b}} \right),$$

which, together with a discussion similar to the one showing (41), implies that, for all $T \geq 1$,

$$\min_{t \in [0:T-1]} \mathbb{E}\left[ \left\| \nabla \hat{f}_{S,\rho}^{\mathrm{SAM}}(\boldsymbol{x}_t) \right\|_2^2 \right] \leq \frac{\mathbb{E}[F_\rho(0)] - f_S^\star}{\sum_{t=0}^{T-1} \eta_t} + 2\sigma C \left( \frac{\rho G}{\sqrt{b}} + \frac{3\sigma}{nb} \sum_{i \in [n]} L_i \right) \frac{T}{\sum_{t=0}^{T-1} \eta_t}$$

$$+ \frac{(|\alpha| + 1)^2 G^2}{n} \sum_{i \in [n]} L_i \left\{ 1 + \frac{4C}{n^2} \left( \sum_{i \in [n]} L_i \right)^2 \right\} \frac{\sum_{t=0}^{T-1} \eta_t^2}{\sum_{t=0}^{T-1} \eta_t}$$

$$+ G\sqrt{4\rho^2 \left(\frac{1}{b^2} + \frac{1}{n^2}\right)\left(\sum_{i\in[n]} L_i\right)^2 + 2\sigma_t^2}$$

$$+ \frac{\rho(|\alpha|+1)G}{n}\sum_{i\in[n]} L_i\left(CG + \frac{B\sigma}{\sqrt{b}}\right).$$

Let $\epsilon > 0$. From (28),

$$\min_{t\in[0:T-1]} \mathbb{E}\left[\left\|\nabla \hat{f}_{S,\rho}^{\mathrm{SAM}}(\boldsymbol{x}_t)\right\|_2^2\right]$$

$$\leq \underbrace{\frac{H_1(\mathbb{E}[F_\rho(0)] - f_S^\star)}{T}}_{V_1 \leq \frac{\epsilon^2}{6}} + \underbrace{\frac{(|\alpha|+1)^2 G^2}{n}\sum_{i\in[n]} L_i\left\{1 + \frac{4C}{n^2}\left(\sum_{i\in[n]} L_i\right)^2\right\}\frac{H_3}{T}}_{V_2 \leq \frac{\epsilon^2}{6}}$$

$$+ \underbrace{2\sigma C H_1\left(\frac{\rho G}{\sqrt{b}} + \frac{3\sigma}{nb}\sum_{i\in[n]} L_i\right)}_{V_3 \leq \frac{\epsilon^2}{6}} + \underbrace{\frac{(|\alpha|+1)^2 G^2 H_2}{n}\sum_{i\in[n]} L_i\left\{1 + \frac{4C}{n^2}\left(\sum_{i\in[n]} L_i\right)^2\right\}}_{V_4 \leq \frac{\epsilon^2}{6}}$$

$$+ \underbrace{\frac{\rho(|\alpha|+1)G}{n}\sum_{i\in[n]} L_i\left(CG + \frac{B\sigma}{\sqrt{b}}\right)}_{V_5 \leq \frac{\epsilon^2}{6}} + \underbrace{G\sqrt{4\rho^2\left(\frac{1}{b^2} + \frac{1}{n^2}\right)\left(\sum_{i\in[n]} L_i\right)^2 + 2\sigma_t^2}}_{V_6 \leq \frac{\epsilon^2}{6}}.$$

There exists $t_2 \in \mathbb{N}$ such that, for all $T \geq t_2$, $V_1 \leq \frac{\epsilon^2}{6}$ and $V_2 \leq \frac{\epsilon^2}{6}$. Moreover, if

$$H_1 \leq \frac{\epsilon^2}{12\sigma C}\left(\frac{\rho G}{\sqrt{b}} + \frac{3\sigma}{nb}\sum_{i\in[n]} L_i\right)^{-1}, \quad (|\alpha|+1)^2 H_2 \leq \frac{n^3 \epsilon^2}{6G^2 \sum_{i\in[n]} L_i\{n^2 + 4C(\sum_{i\in[n]} L_i)^2\}},$$

$$\rho(|\alpha|+1) \leq \frac{n\sqrt{b}\epsilon^2}{6G(\sum_{i\in[n]} L_i)(CG\sqrt{b} + B\sigma)}, \quad \rho^2 \leq \frac{\epsilon^4}{168 G^2}\frac{n^2 b^2}{(n^2 + b^2)(\sum_{i\in[n]} L_i)^2},$$

then $V_i \leq \frac{\epsilon^2}{6}$ ($i = 3, 4, 5, 6$), i.e., (42) holds. $\qquad\square$

*Proof of Theorem 2.4:* Let $\eta_t$ be the cosine-annealing learning rate defined by (15). We then have

$$\sum_{t=0}^{KE-1} \eta_t = \underline{\eta}KE + \frac{\overline{\eta}-\underline{\eta}}{2}KE + \frac{\overline{\eta}-\underline{\eta}}{2}\sum_{t=0}^{KE-1}\cos\left\lfloor\frac{t}{K}\right\rfloor\frac{\pi}{E}.$$

We have

$$\sum_{t=0}^{KE-1}\cos\left\lfloor\frac{t}{K}\right\rfloor\frac{\pi}{E} = \sum_{t=0}^{KE}\cos\left\lfloor\frac{t}{K}\right\rfloor\frac{\pi}{E} - \cos\pi = (K-1) + 1 = K. \tag{44}$$

We thus have

$$\sum_{t=0}^{KE-1}\eta_t = \underline{\eta}KE + \frac{\overline{\eta}-\underline{\eta}}{2}KE + \frac{\overline{\eta}-\underline{\eta}}{2}K$$

$$= \frac{1}{2}\{(\underline{\eta}+\overline{\eta})KE + (\overline{\eta}-\underline{\eta})K\}$$

$$\geq \frac{(\underline{\eta}+\overline{\eta})KE}{2}.$$

Moreover, we have

$$\sum_{t=0}^{KE-1}\eta_t^2 = \underline{\eta}^2 KE + \underline{\eta}(\overline{\eta}-\underline{\eta})\sum_{t=0}^{KE-1}\left(1 + \cos\left\lfloor\frac{t}{K}\right\rfloor\frac{\pi}{E}\right)$$

$$+ \frac{(\overline{\eta} - \underline{\eta})^2}{4} \sum_{t=0}^{KE-1} \left( 1 + \cos \left\lfloor \frac{t}{K} \right\rfloor \frac{\pi}{E} \right)^2,$$

which implies that

$$\sum_{t=0}^{KE-1} \eta_t^2 = \underline{\eta}\overline{\eta}KE + \frac{(\overline{\eta} - \underline{\eta})^2}{4}KE + \underline{\eta}(\overline{\eta} - \underline{\eta}) \sum_{t=0}^{KE-1} \cos \left\lfloor \frac{t}{K} \right\rfloor \frac{\pi}{E}$$

$$+ \frac{(\overline{\eta} - \underline{\eta})^2}{2} \sum_{t=0}^{KE-1} \cos \left\lfloor \frac{t}{K} \right\rfloor \frac{\pi}{E} + \frac{(\overline{\eta} - \underline{\eta})^2}{4} \sum_{t=0}^{KE-1} \cos^2 \left\lfloor \frac{t}{K} \right\rfloor \frac{\pi}{E}.$$

From

$$\sum_{t=0}^{KE} \cos^2 \left\lfloor \frac{t}{K} \right\rfloor \frac{\pi}{E} = \frac{1}{2} \sum_{t=0}^{KE} \left( 1 + \cos 2 \left\lfloor \frac{t}{K} \right\rfloor \frac{\pi}{E} \right)$$

$$= \frac{1}{2}(KE + 1) + \frac{1}{2}$$

$$= \frac{KE}{2} + 1,$$

we have

$$\sum_{t=0}^{KE-1} \cos^2 \left\lfloor \frac{t}{K} \right\rfloor \frac{\pi}{E} = \frac{KE}{2} + 1 - \cos^2 \pi = \frac{KE}{2}.$$

From (44), we have

$$\sum_{t=0}^{KE-1} \eta_t^2 = \frac{(\underline{\eta} + \overline{\eta})^2}{4}KE + \underline{\eta}(\overline{\eta} - \underline{\eta}) + \frac{(\overline{\eta} - \underline{\eta})^2}{2} + \frac{(\overline{\eta} - \underline{\eta})^2}{4}\frac{KE}{2}$$

$$= \frac{3\underline{\eta}^2 + 2\underline{\eta}\overline{\eta} + 3\overline{\eta}^2}{8}KE + \frac{(\overline{\eta} - \underline{\eta})(\overline{\eta} + \underline{\eta})}{2}.$$

Hence, we have

$$\frac{KE}{\sum_{t=0}^{KE-1} \eta_t} \le \frac{2KE}{(\underline{\eta} + \overline{\eta})KE} < \frac{2}{\underline{\eta} + \overline{\eta}} =: H_1$$

and

$$\frac{\sum_{t=0}^{KE-1} \eta_t^2}{\sum_{t=0}^{KE-1} \eta_t} \le \underbrace{\frac{(3\underline{\eta}^2 + 2\underline{\eta}\overline{\eta} + 3\overline{\eta}^2)}{4(\underline{\eta} + \overline{\eta})}}_{H_2} + \frac{1}{KE} \underbrace{(\overline{\eta} - \underline{\eta})}_{H_3}.$$

Accordingly, (28) holds. From (43), we have

$$\frac{2}{\underline{\eta} + \overline{\eta}} \le \frac{\epsilon^2}{12\sigma C} \left( \frac{\rho G}{\sqrt{b}} + \frac{3\sigma}{nb} \sum_{i \in [n]} L_i \right)^{-1},$$

$$(|\alpha| + 1)^2 \frac{(3\underline{\eta}^2 + 2\underline{\eta}\overline{\eta} + 3\overline{\eta}^2)}{4(\underline{\eta} + \overline{\eta})} \le \frac{n^3 \epsilon^2}{6G^2 \sum_{i \in [n]} L_i \{n^2 + 4C(\sum_{i \in [n]} L_i)^2\}}.$$

In particular, when $\underline{\eta} = 0$, we have

$$\frac{2}{\overline{\eta}} \le \frac{\epsilon^2}{12\sigma C} \left( \frac{\rho G}{\sqrt{b}} + \frac{3\sigma}{nb} \sum_{i \in [n]} L_i \right)^{-1},$$

$$(|\alpha| + 1)^2 \frac{3\overline{\eta}}{4} \le \frac{n^3 \epsilon^2}{6G^2 \sum_{i \in [n]} L_i \{n^2 + 4C(\sum_{i \in [n]} L_i)^2\}}.$$

Therefore, Corollary B.1 leads to the assertion in Theorem 2.4.

Let $\eta_t$ be the linear learning rate defined by (16). We then have

$$\sum_{t=0}^{T-1} \eta_t = \overline{\eta}T + \frac{\underline{\eta} - \overline{\eta}}{T} \frac{(T-1)T}{2} = \frac{1}{2}\{(\underline{\eta} + \overline{\eta})T + \overline{\eta} - \underline{\eta}\} > \frac{\underline{\eta} + \overline{\eta}}{2}T,$$

where the third inequality comes from $\overline{\eta} > \underline{\eta}$. We also have

$$\sum_{t=0}^{T-1} \eta_t^2 = \left(\frac{\underline{\eta} - \overline{\eta}}{T}\right)^2 \frac{(T-1)T(2T-1)}{6} + \frac{2(\underline{\eta} - \overline{\eta})\overline{\eta}}{T} \frac{(T-1)T}{2} + \overline{\eta}^2 T$$

$$= \frac{(\underline{\eta} - \overline{\eta})^2(T-1)(2T-1)}{6T} + (\underline{\eta} - \overline{\eta})\overline{\eta}(T-1) + \overline{\eta}^2 T$$

$$< \frac{(\underline{\eta} - \overline{\eta})^2 T}{3} + (\underline{\eta} - \overline{\eta})\overline{\eta}T + \overline{\eta}^2 T$$

$$= \frac{(\underline{\eta} - \overline{\eta})^2 T}{3} + \underline{\eta}\overline{\eta}T$$

$$= \frac{\underline{\eta}^2 + \underline{\eta}\overline{\eta} + \overline{\eta}^2}{3}T,$$

where the third inequality comes from $T - 1 < T$ and $2T - 1 < 2T$. Hence,

$$\frac{T}{\sum_{t=0}^{T-1} \eta_t} < \frac{2}{\underline{\eta} + \overline{\eta}} =: H_1$$

and

$$\frac{\sum_{t=0}^{T-1} \eta_t^2}{\sum_{t=0}^{T-1} \eta_t} < \frac{2(\underline{\eta}^2 + \underline{\eta}\overline{\eta} + \overline{\eta}^2)}{3(\underline{\eta} + \overline{\eta})} =: H_2.$$

Accordingly, (28) holds. From (43), we have that

$$\frac{2}{\underline{\eta} + \overline{\eta}} \leq \frac{\epsilon^2}{12\sigma C} \left(\frac{\rho G}{\sqrt{b}} + \frac{3\sigma}{nb} \sum_{i \in [n]} L_i\right)^{-1},$$

$$(|\alpha| + 1)^2 \frac{2(\underline{\eta}^2 + \underline{\eta}\overline{\eta} + \overline{\eta}^2)}{3(\underline{\eta} + \overline{\eta})} \leq \frac{n^3 \epsilon^2}{6G^2 \sum_{i \in [n]} L_i\{n^2 + 4C(\sum_{i \in [n]} L_i)^2\}}.$$

In particular, when $\underline{\eta} = 0$, we have that

$$\frac{2}{\overline{\eta}} \leq \frac{\epsilon^2}{12\sigma C} \left(\frac{\rho G}{\sqrt{b}} + \frac{3\sigma}{nb} \sum_{i \in [n]} L_i\right)^{-1},$$

$$(|\alpha| + 1)^2 \frac{2\overline{\eta}}{3} \leq \frac{n^3 \epsilon^2}{6G^2 \sum_{i \in [n]} L_i\{n^2 + 4C(\sum_{i \in [n]} L_i)^2\}}.$$

Therefore, Corollary B.1 leads to the assertion in Theorem 2.4. □

## C  TRAINING RESNET-18 ON CIFAR100

The code is available at https://anonymous.4open.science/r/INCREASING-BATCH-SIZE-F09C.
We set $E = 200$, $\eta = \overline{\eta} = 0.1$, and $\underline{\eta} = 0.001$. First, we trained ResNet18 on the CIFAR100
dataset. The parameters, $\alpha = 0.02$ and $\overline{\rho} = 0.05$, used in GSAM were determined by conducting
a grid search of $\alpha \in \{0.01, 0.02, 0.03\}$ and $\rho \in \{0.01, 0.02, 0.03, 0.04, 0.05\}$. Figure 5 compares
the use of an increasing batch size $[16, 32, 64, 128, 256]$ (SGD/SAM/GSAM + increasing_batch)
with the use of a constant batch size 128 (SGD/SAM/GSAM) for a fixed learning rate, 0.1.
SGD/SAM/GSAM + increasing_batch decreased the empirical loss (Figure 5 (Left)) and achieved

higher test accuracies compared with SGD/SAM/GSAM (Figure 5 (Right)). Figure 6 compares the use of a cosine-annealing learning rate defined by (15) (SGD/SAM/GSAM + Cosine) with the use of a constant learning rate, $0.1$ (SGD/SAM/GSAM) for a fixed batch size, $128$. SAM/GSAM + Cosine decreased the empirical loss (Figure 6 (Left)) and achieved higher test accuracies compared with SGD/SAM/GSAM (Figure 6 (Right)).

Table 4 summarizes the mean values of the test errors and the worst-case $\ell_\infty$ adaptive sharpness defined by (Andriushchenko et al., 2023b, (1)) for the parameters $c = (1, 1, \cdots, 1)^\top$ and $\rho = 0.0002$ of the parameter obtained by the algorithm after 200 epochs. SAM+B (SAM + increasing batch) had the highest test accuracy, while GSAM+B (GSAM + increasing_batch) had the lowest sharpness, which implies that GSAM+B approximated a flatter local minimum. The table indicates that using an increasing batch size could avoid sharp local minima to which the algorithms using constant and cosine-annealing learning rates converged.

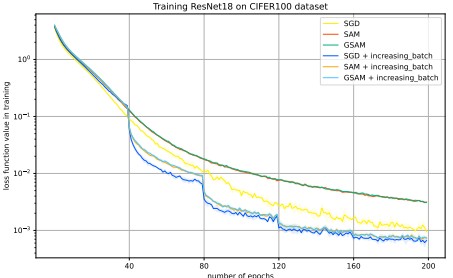
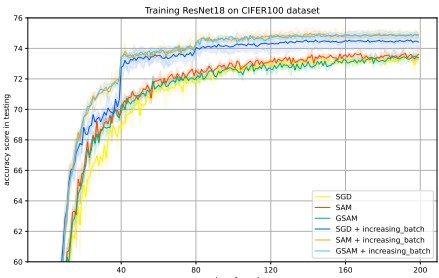

Figure 5: (Left) Loss function value in training and (Right) accuracy score in testing for the algorithms versus the number of epochs in training ResNet18 on the CIFAR100 dataset. The learning rate of each algorithm was fixed at 0.1. In SGD/SAM/GSAM, the batch size was fixed at 128. In SGD/SAM/GSAM + increasing_batch, the batch size was set at 16 for the first 40 epochs and then it was doubled every 40 epochs afterwards, i.e., to 32 for epochs 41-80, 64 for epochs 81-120, 128 for epochs 121 to 160 and 256 for epochs 161 to 200.

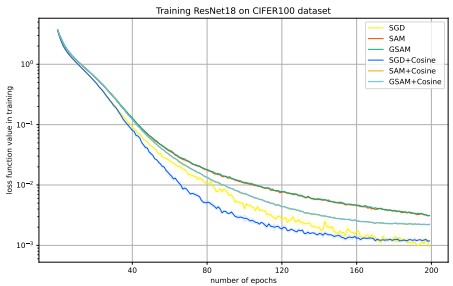
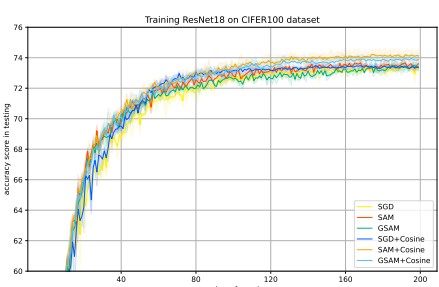

Figure 6: (Left) Loss function value in training and (Right) accuracy score in testing for the algorithms versus the number of epochs in training ResNet18 on the CIFAR100 dataset. The batch size of each algorithm was fixed at 128. In SGD/SAM/GSAM, the constant learning rate was fixed at 0.1. In SGD/SAM/GSAM + Cosine, the maximum learning rate was 0.1 and the minimum learning rate was 0.001.

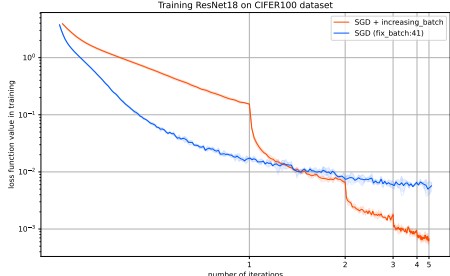 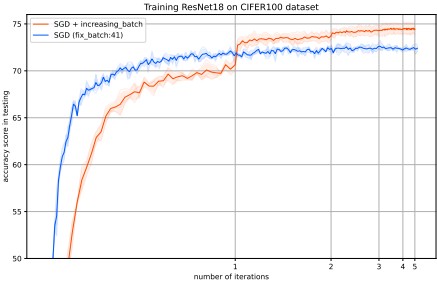

Figure 7: (Left) Loss function value in training and (Right) accuracy score in testing for the batch sizes versus the number of steps in training ResNet18 on the CIFAR100 dataset. The learning rate for each batch size was fixed at 0.1. This is a comparison between the case of a varying batch size [16, 32, 64, 128, 256] (iteration: 242,120) and the case of a fixed batch size of 41 (iteration: 243,800).

Table 4: Mean values of the test errors (Test Error) and the worst-case $\ell_\infty$ adaptive sharpness (Sharpness) for the parameter obtained by the algorithms at 200 epochs of training ResNet18 on the CIFAR100 dataset. "(algorithm)+B" refers to "(algorithm) + increasing batch" in Figure 5, and "(algorithm)+C" refers to "(algorithm) + Cosine" in Figure 6.

|  | SGD | SAM | GSAM | SGD+B | SAM+B | GSAM+B | SGD+C | SAM+C | GSAM+C |
|---|---|---|---|---|---|---|---|---|---|
| Test Error | 26.61 | 26.39 | 26.61 | 25.58 | **25.10** | 25.18 | 26.63 | 25.87 | 26.12 |
| Sharpness | 154.27 | 46.23 | 47.55 | 1.33 | 0.94 | **0.90** | 155.88 | 72.70 | 71.86 |

# D  THE MODEL OF VIT-TINY

|  | Patch size | Embedding Dimension | Heads | Depth | MLP Rate | Params |
|---|---|---|---|---|---|---|
| ViT-Tiny | 4 | 192 | 12 | 9 | 2 | 2.7M |

