# OpenReview forum: "Convergence of Sharpness-Aware Minimization Algorithms using Increasing Batch Size and Decaying Learning Rate"
_ICLR.cc/2025/Conference — ICLR 2025 Conference Withdrawn Submission_

### Official Review · Reviewer_K6Bk · 2024-11-02

**Soundness:** 2
**Presentation:** 1
**Contribution:** 2
**Rating:** 3
**Confidence:** 4

**Summary:**

- The authors consider empirically and theoretically the GSAM algorithm with various learning rates, batch sizes, and decay learning rates techniques.
- The authors compare GSAM in different settings and empirically analyze the convergence rate and the loss landscape when using increasing large batch size and decay algorithms.

**Strengths:**

- This work progressively establishes and justifies its framework
- The results are promising, however, I have some concerns regarding the results as discussed below.

**Weaknesses:**

- One of the main weaknesses of this paper comes from the significance of the paper. To be more specific, the algorithm seems to be incremental and the contribution lies in fine-tuning GSAM in different settings. Note that this fine-tuning is compulsory for any algorithm to
- Although the authors evaluate the algorithms on different model architectures. It would be better if the authors evaluated the algorithms on more datasets to prove the effectiveness of the proposed algorithm in various settings.
- The writing seems to be verbose, especially in the introduction, and thus, does not make the reader understand clearly what are the issues of current works, what is the motivation that leads to the paper’s contribution.
- The paper requires more proofreading as some parts of the paper are not easy to follow and lack coherence. For examples:
    - P1/015 - P1/01.
    - P1/030 - P1/033.
    - Perturbed empirical loss and maximum empirical loss needed to be discussed in more detailedly as these terminologies are not widely used in many AI/ML works.
    - P4/187 - P4/196 is verbose, it is hard to understand what information the authors want to convey. The Eq.~(8) is trivial and does not bring a new observation to readers.
    - P4/213 - P5/217.
    - Is Theorem 2.3 also Theorem B.1?
- Despite the paper shows potential, many issues need to be addressed.

**Questions:**

1. Can the author explain more about the main concern stated in P1/030 - P1/032? The concern sounds trivial and has been widely considered in many works. For example:
- [R1]
- [R2]
2. What do the authors mean in P1/35 - P1/37: “which may lead to better generalization than finding sharp minima”? From my limited understanding, the two attached researches are not about finding the sharp minima. Furthermore, finding sharp minima seems not to be the actual contribution and research direction in the topics of improving generalization.
3. In P1/043 - P1/048: The two sentences mentioned by the authors are conflicts. For instance, while the first sentence said that using large batch size falls into sharp local minima, the second sentence said that increasing batch size improves the generalization. What is the information that the authors want to convey?
4. Where is the Approximated SAM problem in the paper [R1]?
5. Can you explain why (3) in the research is equivalent to (2) in [R1]?
6. Can you explain more about the statements about GSAM in P3/158-161: “GSAM using b loss function which are randomly chosen at time t”?
7. What does the author mean by $\xi$ as a random variable in P4/166? What does it operate in Assumption 2.1?
8. The replacement of S and $S_t$ in (6) is trivial and should be truncated or put in the appendix to save space for more important information.
9. In P4/187 - P4/196, after mentioning SAM update, why do we talk about GSAM? What is the relationship between GSAM and SAM in this case? What is the key takeaway?
10. Moreover, despite Section 2.2 being a Mini-batch GSAM algorithm, we do not see much about the key features of GSAM, we only see the SAM algorithm derivations.
11. Is algorithm 1 a contribution to the paper?
12. The upper bound in L794-L795 is too large, I worry that the too large upper-bound may make every equation in the proof become too high. Thus, making (21) to (22) become trivial. As a consequence, Theorem 2.1 becomes trivial. Not that the L value may be very large.
13. Similarly, the lower bound seems to be loose (L864 - L878). Please discuss this.
14. Can the authors explain the significance of Theorem 2-3 in the convergence analysis?
15. Is $\epsilon$ in L323 also the $\epsilon$ in (2). If so, the convergence rate of GSAM is fixed over the training, isn’t it? A similar question applied to Theorem 2.4.
16. Did the authors conduct empirical studies with different weight initialization? Or can the authors provide any theoretical study to prove the finding observation?
17. Did the authors conduct an ablation test on the combination of methods with different kinds of weight initialization mechanisms?

Reference:
- [R1]: Pierre Foret, Ariel Kleiner, Hossein Mobahi, Behnam Neyshabur, Sharpness-aware Minimization for Efficiently Improving Generalization, ICLR 2021.
- [R2]: Maksym Andriushchenko, Nicolas Flammarion, Towards Understanding Sharpness-Aware Minimization, Towards Understanding Sharpness-Aware Minimization, ICML 2022.
- [R3]: Yong Liu, Siqi Mai, Xiangning Chen, Cho-Jui Hsieh, Yang You, Towards Efficient and Scalable Sharpness-Aware Minimization, CVPR 2022.

---

### Official Review · Reviewer_mzwx · 2024-11-03

**Soundness:** 2
**Presentation:** 3
**Contribution:** 2
**Rating:** 5
**Confidence:** 3

**Summary:**

This paper investigates the ideas of increasing batch size and decaying learning rate for GSAM. The paper mainly provides convergence analyses for both cases showing that GSAM can achieve $\epsilon$-accuracy with these two strategies. The paper also provide numerical evaluations on some image classification tasks to support their theoretical results.

**Strengths:**

- The upper and lower bounds for the difference between GSAM and GD seem to be reasonable, achieving that GSAM can behave similarly to GD under the two strategies of this paper's interest.
- The convergence results indicating the existence of t yielding within $\epsilon$-accuracy are interesting.

**Weaknesses:**

- The theory results focused on showing the $\epsilon$ convergence are a bit nonrealistic, requiring asymptotically increasing batch size to the full batch for example. This creates a gap between theory and practical settings, diminishing the significance of the paper. Also, in the extremely simplified settings with increasing batch size for GD, this results are quite expected.
- "Increasing batch size" used in theory and its implementation for experiments are different.
- The numerical results only cover for CIFAR 100 which is too small and may not generalize, diminishing the empirical validation. ViT results are quite weak and do not support well the theory results.
- The choice of a particular class of SAM (GSAM) is somewhat arbitrary, possibly favoring analyses and potentially limiting the generality of the results; in experiments, it appears that this choice turns to be quite random as well because GSAM is not consistently better than SAM both in terms of either accuracy or flatness.

**Questions:**

- Seeing decreased flatness (for ResNet) is interesting. Is this result interpreted in the same manner as in the convergence or can authors suggest different ways to understand this result?

---

### Official Review · Reviewer_BJXC · 2024-11-03

**Soundness:** 2
**Presentation:** 2
**Contribution:** 2
**Rating:** 3
**Confidence:** 2

**Summary:**

This paper analyzes the convergence of GSAM, a variant of SAM optimizer, with an increasing batch and constant learning rate, also with a constant batch size and decaying learning rate. Numeric experiments partially validate their analyses.

**Strengths:**

1. The theoretical analysis looks sound. The setting of increasing batch size is interesting to investigate, which is a good perspective.
2. The experiments are extensive. Seemingly increasing batch size during training indeed avoids sharp minima in practice and leads to a better generalization ability. This message is insightful.

**Weaknesses:**

1. The contribution of this paper is seemingly marginal and incremental. I am not an expert in convergence analysis, but as shown in the table 1, the convergence results of SAM and its variants has been well established before. This paper focuses on a variant of SAM, namely GSAM, and its convergence has been established before under a different setting. The authors are suggested to discuss more on the significance of this problem.
2. The setting of this paper is motivated by the observations that increasing the batch size or decaying the learning rate avoids sharp local minima of the empirical loss, however, the convergence analysis under this setup cannot explain the phenomenon of avoiding sharp local minima. The theoretical insights drawn from its analyses are limited.
3. In Tab. 2, "SAM+C" and "GSAM+C" shows no better flatness compared to vanilla SAM or GSAM. This observation contradicts with previous observation that increasing the batch size or decaying the learning rate avoids sharp local minima of the empirical loss.
4. The authors provide experiments in real-world scenarios, however, some synthetic experiments are also encouraged. Because experiments under synthetic setup could better validate the correctness of the theorems, while real-world experiments are hard to control variables.

**Questions:**

1. In Fig 1 and Fig 2., SAM with a fixed batch size or constant learning rate shows no better test accuracy compared with SGD in the same setting. However, in practice, it is expected that SAM outperforms SGD in the same setting. Is there any explanation for this phenomenon?

---

### Official Review · Reviewer_jF3B · 2024-11-04

**Soundness:** 2
**Presentation:** 2
**Contribution:** 1
**Rating:** 3
**Confidence:** 4

**Summary:**

Paper addressen generalization and convergence properties of Deep learning. In particular, incsiperd by previous works, it takes SAM and applies hypothesis of "increasing batch sizes or decaying learning rates leading to flatter optima" to Aproximated SAM problem Problem 2.1. In a series of Theorems and corroborated by experimetns on ResNet, ViT on CIFAR the claims of the hypothesis are argued to hold.

**Strengths:**

S1: Well chosen and interesting topic

**Weaknesses:**

W1: To me, experiments seem to express a faster convergence for batch size progression $\{8,16,32,64,128\}$ compared to just $128$ batch size training. In other words, the 'early stopping' regularisation effect, which is well-known and widely used. It is clearly demonstrated in Fig.1 when at epoch 40 (first batch increase, I believe) the training and error of small batch models outperform already the large batch size models. Similar in Figure 2, 3, 4 and so on. Thus one can propose that instead of showing the merits of decaying learning rate or increasing batch size, what is reported is the opposite - a faster convergence of larger learning rate or smaller batch size.

I find provided experiments not convincing and I would encourage authors to argue otherwise. Until then it seems to me extremely important authors providing a much stronger evidence for the claims of the paper on carefully chosen experimental setting, before paper could be considered for accepting.

W1a. Labels and text used in Figures render those hardly readable and surely not sufficient for a potential camera ready version. An obvious update is suggested.

Theoretical part:
W2: My concerns on too many approximations and indications and assumptions taken and combined are summarized in line 379: "... Theorem 2.4 indicates that the parameters |α| and ρ in (18) become small and thereby achieve an ε–approximation of GSAM, ..." - The presented theory is an approximation of GSAM that is an approximation to SAM that is an approximation to exact min-max loss. On top is quite hard to verify it as paper ofter refer to Theorems, equations and algorithm form other papers etc. and take many assumtions ... Theo retical argument in the paper seems to be suitable for an 'experimental paper' rather than rigorous theory paper. Then it is even more important to produce good experiments ... see W1.

W2b: Convergence Theorems 2.3, 2.4 claim converegce result of the $expected$ gradient. What does it really say about individial training paths? Can we even conclude anything without bounds on higher moments or likes?


W4: In conclusion, l 537-539: "... The numerical results showed that, compared with SGD/Adam, SAM/GSAM with an increasing batch size and a constant learning rate converges to flatter local minima of the empirical loss functions ..." - I can't follow how experiments showed any convergence to $flatter$ local minima? It was only demonstrated (unconvincingly, see other comments) that test loss after early stopping is lower. Could authors elaborate or adjust?


Overall and despite authors claims, to me paper seems more experimental than theoretical work (see weaknesses above). Yet, experimental results are unconvincing (see comments) for given, rather strong claims of the paper. Extensive update and additional experiments would be required before my recommending the paper for acceptance. I encourage authors to addres raised concernes and I would more than willing to raise my score, however.

**Questions:**

For major question please respond to Weaknesses section above. On top:

Q1: l 45-47: Is the claim fully correct? For instance referred work (Smith et al., 2018) rather argues for merits of increasing batch size over learning rate annealing, comaring those relatively, not cllaiming that both of each produce flatter otima solutions compared to fixed learning rate. Could authors clarify please?

Q2: How is i.i.d. A3 assumtion (Assumtion 2.1) of losses achived in practice/experiments? Would not be losses dependent due to shared weight updates over the course of training?

---

### Note · Authors · 2024-12-05

I have read and agree with the venue's withdrawal policy on behalf of myself and my co-authors.